# A-FedPD: Aligning Dual-Drift is All Federated Primal-Dual Learning Needs

**Yan Sun**
The University of Sydney
ysun9899@uni.sydney.edu.au

**Li Shen**[*]
Shenzhen Campus of Sun Yat-sen University
mathshenli@gmail.com

**Dacheng Tao**[†]
Nanyang Technological University
dacheng.tao@ntu.edu.sg

## Abstract

As a popular paradigm for juggling data privacy and collaborative training, federated learning (FL) is flourishing to distributively process the large scale of heterogeneous datasets on edged clients. Due to bandwidth limitations and security considerations, it ingeniously splits the original problem into multiple subproblems to be solved in parallel, which empowers *primal dual* solutions to great application values in FL. In this paper, we review the recent development of classical *federated primal dual* methods and point out a serious common defect of such methods in non-convex scenarios, which we say is a "dual drift" caused by dual hysteresis of those longstanding inactive clients under partial participation training. To further address this problem, we propose a novel ***Aligned Federated Primal Dual*** (***A-FedPD***) method, which constructs virtual dual updates to align global consensus and local dual variables for those protracted unparticipated local clients. Meanwhile, we provide a comprehensive analysis of the optimization and generalization efficiency for the *A-FedPD* method on smooth non-convex objectives, which confirms its high efficiency and practicality. Extensive experiments are conducted on several classical FL setups to validate the effectiveness of our proposed method.

## 1 Introduction

Since McMahan et al. [2017] propose the *federated average* paradigm, FL has gradually become a promising approach to handle both data privacy and efficient training on the large scale of edged clients, which employs a global server to coordinate local clients jointly train one model. Due to privacy protection, it disables the direct information interaction across clients. All clients must only communicate with an accredited global server. This paradigm creates an unavoidable issue, that is, bandwidth congestion caused by mass communication. Therefore, FL advocates training models on local clients as much as possible within the maximum bandwidth utilization range and only communicates with a part of clients per communication round in a partial participation manner. Under this particular training mechanism, FL needs to effectively split the original problem into several subproblems for local clients to solve in parallel. Because of this harsh limitation, general algorithms are often less efficient in practice. But the *primal dual* methods just match this training pattern, which empowers it with huge application potential and great values in FL.

---

[*]Li Shen is the corresponding author.
[†]Dr Tao's research is partially supported by NTU RSR and Start Up Grants.

38th Conference on Neural Information Processing Systems (NeurIPS 2024).

*Primal dual* methods, which are specified as *Lagrangian primal dual* in this paper, solve the problem by penalizing and relaxing the constraints to the original objective via non-negative Lagrange multipliers, which make great progress in convex optimization. It benefits from the consideration of splitting a large problem into several small simple problems to solve, which has been widely developed and applied in the distributed framework as a global consensus problem. This solution is also well suited to the FL scenarios for its effective split characteristic. Recent studies revealed the application potential of such methods. Since Tran Dinh et al. [2021] propose the *Randomized Douglas-Rachford Splitting* in FL, which unblocks the study of this important branch. With further exploration of Zhang et al. [2021], Durmus et al. [2021], Zhang and Hong [2021], *federated primal dual* methods are proven to achieve the fast $\mathcal{O}(1/T)$ convergence rate. Then it is expanded to the more complicated scenarios and incorporated with several novel techniques to achieve state-of-the-art (SOTA) performance in the FL community, which further confirms the great contributions.

However, as studies go further, a series of problems of *federated primal dual* methods in the experiments are also exposed. Sensitivity to hyperparameters and fluctuations affected by the large scale makes it extremely unstable in practice, especially in the partial participation manner which is one of the most important concerns in FL. Specifically, *primal dual* methods successfully solve the problems by alternately updating each primal variable and each dual variable. When it is grafted onto the partial participation training in FL, most clients will remain inactive for a long time, which means most of the dual variables will be stagnant and very outdated in the training. As the training process continues, when one long-term inactive client is reactivated

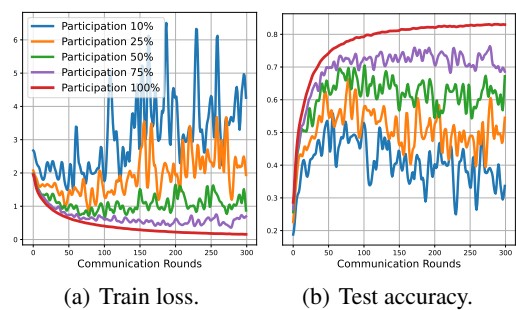

(a) Train loss.    (b) Test accuracy.

Figure 1: "*Dual drift*" issue of the *federated primal dual* method under different participation ratios. When the participation ratio is low, *dual drift* introduces a very large variance, yielding divergence.

to participate in training, the solving process of its local subproblem will become extremely unstable due to excessive differences between the primal and dual variables, and sometimes even fail. We call this a "*dual drift*" problem, which is also one of the most formidable challenges in practice in FL. In Fig.1, we clearly show how the "*dual drift*" deteriorates as the participation ratio decreases.

To efficiently expand the *primal dual* methods to partial participation scenarios while enhancing the training stability in practice, and further alleviate the "*dual drift*" problem, we propose a novel algorithm, named *Aligned **Fed**erated **P**rimal **D**ual* (*A-FedPD*), which constructs the virtual dual updates for those unparticipated clients per communication round to align with primal variables. Concretely, after each communication round, we first aggregate the local solutions received from active clients as the unbiased approximation of the local solution of those unparticipated clients. Then we provide a virtual update on the dual variables to align with the primal variable in the training. Updating errors for dual variables will be diminished as global consensus is achieved. The proposed *A-FedPD* method enables unparticipated clients to keep up-to-date, which approximates the quasi-full-participation training, which can efficiently alleviate the "*dual drift*" in practice.

Furthermore, we provide a comprehensive analysis of the optimization and generalization efficiency of the proposed *A-FedPD* method, which also could be easily extended to the whole *federated primal dual family*. On smooth non-convex objectives, compared with the vanilla *FedAvg* method, the *A-FedPD* could achieve a fast $\mathcal{O}(1/T)$ convergence rate which maintains consistent with SOTA *federated primal dual* methods. Moreover, it could support longer local training without affecting stability. Under the same training costs, the *A-FedPD* method achieves less generalization error. We conduct extensive experiments to validate its efficiency across several general federated setups. We also test a simple variant to show its good scalability incorporated with other novel techniques in FL.

We summarize our major contributions as follows:

- We review the development of *federated primal dual family* and point out one of its most formidable challenges in the practical application in FL, which is summarized as the "*dual drift*" problem in this paper.

- We propose a novel *A-FedPD* method to alleviate the "*dual drift*", which constructs the virtual update for dual variables of those unparticipated clients to align with primal variables.

- We provide a comprehensive analysis of the optimization and generalization efficiency of *A-FedPD*. It could achieve a fast convergence rate and a lower generalization error bound than the vanilla *FedAvg* method.

- Extensive experiments are conducted to validate the performance of the *A-FedPD* method. Furthermore, we test a simple variant of it to show its good scalability.

## 2 Related Work

*Federated primal average.* Since McMahan et al. [2017] propose the *FedAvg* paradigm, a lot of *primal average*-based methods are learned to enhance its performance. Most of them target strengthening global consistency and alleviating the "client drift" problem [Karimireddy et al., 2020]. Li et al. [2020] propose to adopt proxy terms to control local updates. Karimireddy et al. [2020] propose the variance reduction version to handle the biases in the *primal average*. Similar implementations include [Dieuleveut et al., 2021, Jhunjhunwala et al., 2022]. Moreover, momentum-based methods are also popular for correcting local biases. Ozfatura et al. [2021], Xu et al. [2021] propose to adopt the global consistency controller in the local training to force a high consensus. Remedios et al. [2020] expand the local momentum to achieve higher accuracy in the training. Similarly, Wang et al. [2019], Kim et al. [2022] incorporate the global momentum which could further improve its performance. Wang et al. [2020b] tackle the local inconsistency and utilize the *weighted primal average* to balance different clients with different computing power. Based on this, Horváth et al. [2022] further select the important clients set to balance the training trends under different heterogeneous datasets. Liu et al. [2023] summarize the inertial momentum implementation which could achieve a more stable result. Qu et al. [2022] utilize the *Sharpeness Aware Minimization* (*SAM*) [Foret et al., 2020] to make the loss landscape smooth with higher generalization performance. Then Caldarola et al. [2022, 2023] propose to improve its stability via *Adaptive SAM* and window-based model averaging. In summary, *Federated primal average* methods focus on alleviating the local inconsistency caused by "client drift" [Malinovskiy et al., 2020, Wang et al., 2020a, Charles and Konečnỳ, 2021]. However, as analyzed by Durmus et al. [2021], the *federated primal dual* methods will regularize the local objective gradually close to global consensus by dynamically adjusting the dual variables. This allows "client drift" to be effectively translated into a dual consistency problem on cross-silo devices.

*Convex optimization of federated primal dual.* The primal-dual method was originally proposed to solve convex optimization problems and achieved high theoretical performance. In the FL setups, this method has also made significant advancements. Grudzień et al. [2023a] compute inexactly a proximity operator to work as a variant of primal dual methods. Another technique is loopless instead of an inner loop of local steps. Mishchenko et al. [2022] propose the *Scaffnew* method to achieve the higher optimization efficiency, which is interpreted as a variant of primal dual approach [Condat and Richtárik, 2022]. These techniques can also be easily combined with existing efficient communication methods, e.g. compression and quantization [Grudzień et al., 2023b, Condat et al., 2023].

*Non-convex optimization of federated primal dual.* Since Tran Dinh et al. [2021] adopt the *Randomized Douglas-Rachford Splitting*, which unblocks the study of the important branch of utilizing *primal dual* methods in FL [Pathak and Wainwright, 2020]. With further exploration of Zhang et al. [2021], *federated primal dual* methods are proven to achieve the fast convergence rate. Yuan et al. [2021] learn the composite optimization via a *primal dual* method in FL. Meanwhile, Sarcheshmehpour et al. [2021] empower its potential on the undirected empirical graph. Shen et al. [2021] also study an agnostic approach under class imbalance targets. Then, Gong et al. [2022], Wang et al. [2022] expand its theoretical analysis to the partial participation scenarios with global regularization. Durmus et al. [2021] improve its implementation by introducing the global dual variable. Moreover, Zhou and Li [2023] learn the effect of subproblem precision on training efficiency. Sun et al. [2023b,a] incorporate it with the *SAM* to achieve a higher generalization efficiency. Niu and Wei [2023] propose hybrid *primal dual* updates with both first-order and second-order optimization. Wang et al. [2023] propose a variance reduction variant to further improve the training efficiency. Li et al. [2023a] expand it to a decentralized approach, which could achieve comparable performance in the centralized. Tyou et al. [2023] propose a localized *primal dual* approach for FL training. Li et al. [2023b] further explore its efficiency on the specific non-convex objectives with non-smooth regularization. Current researches reveal the great application value of the *primal dual* methods in FL. However, most of them still face the serious "*dual drift*" problem at low participation ratios. Our improvements further alleviate this issue and make the *federated primal dual* methods perform more stably in the FL community.

# 3 Methodology

We first review the *primal dual* methods in FL and demonstrate the "*dual drift*" issue. Then, we demonstrate the *A-FedPD* approach to eliminate the "*dual drift*" challenge. Notations are stated in Table 1. Other symbols are defined when they are first introduced. We denote $\mathbb{R}$ as the real set and $\mathbb{E}$ as the expectation in the corresponding probability space. Other notations are defined as first stated.

Table 1: Notations adopted in this paper.

| Symbol | Notations |
|---|---|
| $\mathcal{C}$ / $C$ | client set / size of client set |
| $\mathcal{P}$ / $P$ | active client set / size of active client set |
| $\mathcal{S}_i$ / $S$ | local dataset / size of local dataset |
| $\theta$ / $\theta_i$ | global parameters / local parameters |
| $\lambda$ / $\lambda_i$ | global dual parameters / local dual parameters |
| $T$ / $t$ | training round / index of training round |
| $K$ / $k$ | local interval / index of local interval |
| $\rho$ | proxy coefficient |

## 3.1 Preliminaries

**Setups.** In the general and classical federated frameworks, we usually consider the general objective as a finite-sum minimization problem $F(\theta) : \mathbb{R}^d \to \mathbb{R}$,

$$\min_{\theta} F(\theta) = \frac{1}{C} \sum_{i \in \mathcal{C}} F_i(\theta), \quad F_i(\theta) \triangleq \mathbb{E}_{\xi \sim \mathcal{D}_i} f_i(\theta, \xi). \tag{1}$$

In each client, there exists a local private data set $\mathcal{S}_i$ which is considered a uniform sampling set of the distribution $\mathcal{D}_i$. In FL setups, $\mathcal{D}_i$ is unknown to others for the privacy protection mechanism. Therefore, we usually consider the local *Empirical Risk Minimization (ERM)* as:

$$\min_{\theta} f(\theta) = \frac{1}{C} \sum_{i \in \mathcal{C}} f_i(\theta), \quad f_i(\theta) \triangleq \frac{1}{S} \sum_{\xi \in \mathcal{S}_i} f_i(\theta, \xi). \tag{2}$$

Our desired optimal solution is $\theta^\star = \arg\min F(\theta)$. However, we can only approximate it on the limited dataset as $\theta_{\mathcal{S}}^\star = \arg\min f(\theta)$, which spontaneously introduces the unavoidable biases on its generalization performance. This is one of the main concerns in the field of the current machine learning community. Motivated by this imminent challenge, we conduct a comprehensive study on the performance of *primal dual*-based algorithms in FL and further propose an improvement to enhance its generalization efficiency and stability performance.

## 3.2 *Primal Dual*-family in FL

*Primal Dual* methods optimize the global objective by decomposing it into several subproblems and iteratively updating local variables incorporated by Lagrangian multipliers [Boyd et al., 2011], which gives it a unique position in solving FL problems. Due to local data privacy, we have to split the global task into several local tasks for optimization on their private dataset. This similarity also provides an adequate foundation for their applications in the FL community. A lot of studies extend it to the general FL framework and achieve considerable success.

We follow studies [Zhang et al., 2021, Durmus et al., 2021, Wang et al., 2022, Gong et al., 2022, Sun et al., 2023b, Zhou and Li, 2023, Sun et al., 2023a, Fan et al., 2023, Zhang et al., 2024] to summarize the original objective Eq.(2) as the global consensus reformulation:

$$\min_{\theta, \theta_i} \frac{1}{C} \sum_{i \in \mathcal{C}} f_i(\theta_i), \quad \text{s.t.} \quad \theta_i = \theta, \quad \forall i \in \mathcal{C}. \tag{3}$$

By relaxing equality constraints $\theta_i = \theta$, Eq.(3) is separable across different local clients. Wang et al. [2022] demonstrate the difference between the solution on the primal problem and dual problem in detail and confirm the equivalence of these two cases in FL. By penalizing the constraint on the local objective $f_i$, we can define the augmented Lagrangian function associated with Eq.(3) as:

$$\mathcal{L}(\theta_i, \theta, \lambda_i) = \frac{1}{C} \sum_{i \in \mathcal{C}} \left[ f_i(\theta_i) + \langle \lambda_i, \theta_i - \theta \rangle + \frac{\rho}{2} \|\theta_i - \theta\|^2 \right], \tag{4}$$

where $\rho$ denotes the penalty coefficient. To train the global model, each local client should first minimize the local augmented Lagrangian function and solve for local parameters. Based on updated local parameters, we then update the dual variable to align the Lagrangian function with the consensus

constraints. Finally, we minimize the augmented Lagrangian function and solve for the consensus. Objective Eq.(3) could be solved after multiple alternating updates as:

$$
\begin{cases}
\theta_i^{t+1} & = & \arg\min_{\theta_i} \mathcal{L}(\theta_i^t, \theta^t, \lambda_i^t) \quad i \in \mathcal{C}, \\
\lambda_i^{t+1} & = & \lambda_i^t + \rho(\theta_i^{t+1} - \theta^t), \\
\theta^{t+1} & = & \frac{1}{C} \sum_{i \in \mathcal{C}} (\theta_i^{t+1} + \frac{1}{\rho} \lambda_i^{t+1}).
\end{cases}
\tag{5}
$$

We then review the classical *federated primal dual* methods.

**FedPD**. Zhang et al. [2021] propose a general federated framework from the primal-dual optimization perspective which can be directly summarized as Eq.(5). As an underlying method in the federated *primal dual*-family, it requires all local clients to participate in the training per round, which also significantly reduces communication efficiency.

**FedADMM**. Wang et al. [2022] extend the theory of the primal-dual optimization in FL and prove the equivalence between *FedDR* [Tran Dinh et al., 2021] and *FedADMM*. Furthermore, it considers the complete format of composite objective $f(\theta_i) + g(\theta)$. To optimize the composite objective, after the iterations of Eq.(5), it additionally solves the proximal step on the function $g(\theta)$:

$$
\begin{cases}
\theta_i^{t+1} & = & \arg\min_{\theta_i} \mathcal{L}(\theta_i^t, \theta^t, \lambda_i^t) + g(\theta^t) \quad i \in \mathcal{P}^t, \\
\lambda_i^{t+1} & = & \lambda_i^t + \rho(\theta_i^{t+1} - \theta^t), \\
\overline{\theta}^{t+1} & = & \frac{1}{P} \sum_{i \in \mathcal{P}^t} (\theta_i^{t+1} + \frac{1}{\rho} \lambda_i^{t+1}), \\
\theta^{t+1} & = & \arg\min g(\theta^t) + \frac{1}{2\rho} \|\theta^t - \overline{\theta}^{t+1}\|^2.
\end{cases}
\tag{6}
$$

*FedADMM* introduces a more general update with the regularization term $g(\theta)$ and supports the partial participation training mechanism, which also brings a great application value of primal-dual methods to the FL community. When $g(\cdot) \equiv 0$, it degrades to the partial *FedPD* by $\theta^{t+1} = \overline{\theta}^{t+1}$. When $\mathcal{P}^t \neq \mathcal{C}$, "*dual drift*" brings great distress for training.

**FedDyn**. Durmus et al. [2021] utilize the insight of primal-dual optimization to introduce a dynamic regularization term to solve the local augmented Lagrangian function, which is actually the dual variable in *FedADMM*. Differently, they propose a global dual variable $\lambda^t$ to update global parameters $\theta^t$ instead of only active local dual variables $\lambda_i^t$ ($i \in \mathcal{P}^t$):

$$
\begin{cases}
\theta_i^{t+1} & = & \arg\min_{\theta_i} \mathcal{L}(\theta_i^t, \theta^t, \lambda_i^t) \quad i \in \mathcal{P}^t, \\
\lambda_i^{t+1} & = & \lambda_i^t + \rho(\theta_i^{t+1} - \theta^t), \\
\lambda^{t+1} & = & \lambda^t + \rho \frac{1}{C} \sum_{i \in \mathcal{P}^t} (\theta_i^{t+1} - \theta^t), \\
\theta^{t+1} & = & \frac{1}{P} \sum_{i \in \mathcal{P}^t} \theta_i^{t+1} + \frac{1}{\rho} \lambda^t.
\end{cases}
\tag{7}
$$

Compared with *FedADMM*, although the global dual variable further corrects the primal parameters, it still hinders the training efficiency, which must rely on the anachronistic historical directions of the local dual variables. Moreover, the global dual variable always updates slowly, which results in consensus constraints that are more difficult to satisfy when solving local subproblems.

**Dual drift**. Kang et al. [2024] have indicated that the update mismatch between primal and dual variables leads to a "drift". Here, we provide a detailed analysis of the key differences caused by this mismatch. When adopting partial participation, each client is activated at a very low probability, especially on a large scale of edged devices, which widely leads to very high hysteresis between global parameters $\theta$ and local dual variable $\lambda_i$. For instance, at round $t$, we select a subset $\mathcal{P}^t$ to participate in current training and then update the global parameters by $\theta^{t+1} = \arg\min_\theta \mathcal{L}(\theta_i^{t+1}, \theta^t, \lambda_i^t)$ for $i \in \mathcal{P}^t$. Then at round $t+1$, when a client $i \notin \{\mathcal{P}^\tau\}_{\tau=t_0+1}^t$ ($t_0 \ll t$) that has not been involved in training for a long time is activated, its local dual variable $\lambda_i^{t_0}$ may severely mismatch the current global parameters $\theta^t$. This triggers that the local subproblem $\mathcal{L}(\theta_i^{t+1}, \theta^{t+1}, \lambda_i^{t_0})$ fail to be optimized properly and even become completely distorted in extreme scenarios, yielding a "*dual drift*" issue.

### 3.3 A-FedPD Method

As introduced in the last part, "*dual drift*" problem usually results in the unstable optimization of each local subproblem under partial participation. To further mitigate the negative effects of *dual drift* problems and improve the training efficiency, we propose a novel *A-FedPD* method (see Algorithm 1), which aligns the virtual dual variables of unparticipated clients via global average models.

**Algorithm 1** A-FedPD Algorithm

**Input:** $\theta^0, \theta_i^0, T, K, \lambda_i^0, \rho$
**Output:** global average model
1: **Initialization** : $\theta_i^0 = \theta^0, \lambda_i^0 = 0$.
2: **for** $t = 0, 1, 2, \cdots, T - 1$ **do**
3:     randomly select active clients set $\mathcal{P}^t$ from $\mathcal{C}$
4:     **for** client $i \in \mathcal{P}^t$ **in parallel do**
5:         receive $\lambda_i^t, \theta^t$ from the global server
6:         $\theta_i^{t+1} = LocalTrain(\lambda_i^t, \theta^t, \eta^t, K)$
7:         send $\theta_i^{t+1}$ to the global server
8:     **end for**
9:     $\overline{\theta}^{t+1} = \frac{1}{P} \sum_{i \in \mathcal{P}^t} \theta_i^{t+1}$
10:    $\lambda_i^{t+1} = D\text{-}Update(\lambda_i^t, \theta^t, \theta_i^{t+1}, \overline{\theta}^{t+1}, \mathcal{P}^t)$
11:    $\overline{\lambda}^{t+1} = \frac{1}{C} \sum_{i \in \mathcal{C}} \lambda_i^{t+1}$
12:    $\theta^{t+1} = \overline{\theta}^{t+1} + \frac{1}{\rho} \overline{\lambda}^{t+1}$
13: **end for**
14: return global average model

$\diamond$ *LocalTrain*: (Optimize Eq.(4))
**Input:** $\lambda_i^t, \theta^t, \eta^t, K$
**Output:** $\theta_{i,K}^t$
1: **for** $k = 0, 1, 2, \cdots, K - 1$ **do**
2:     calculate the stochastic gradient $g_{i,k}^t$
3:     $\theta_{i,k+1}^t = \theta_{i,k}^t - \eta^t(g_{i,k}^t + \lambda_i^t + \rho(\theta_{i,k}^t - \theta^t))$
4: **end for**

$\diamond$ *D-Update*: (update dual variables)
**Input:** $\lambda_i^t, \theta^t, \theta_i^{t+1}, \overline{\theta}^{t+1}, \mathcal{P}^t$
**Output:** $\lambda_i^{t+1}$
1: **if** $i \in \mathcal{P}^t$ **then**
2:     $\lambda_i^{t+1} = \lambda_i^t + \rho_t(\theta_i^{t+1} - \theta^t)$
3: **else**
4:     $\lambda_i^{t+1} = \lambda_i^t + \rho_t(\overline{\theta}^{t+1} - \theta^t)$
5: **end if**

Specifically, we solve dual variables for unified management and distribution. At round $t$, we select an active client set $\mathcal{P}^t$ and send the corresponding variables to each active client. Then local client solves the subproblem with $K$ stochastic gradient descent steps and sends the last state $\theta_{i,K}^t$ back to the global server. On the global server, it first aggregates the updated parameters as $\overline{\theta}^{t+1}$. Then it performs the updates of the dual variables. For each active client $i \in \mathcal{P}^t$, it equally updates the local dual variable as vanilla *FedPD*. For the unparticipated clients $i \notin \mathcal{P}^t$, they update the virtual dual variable with the aggregated parameters $\overline{\theta}^{t+1}$. Finally, we can update the global model with the aggregated parameters and averaged dual variables. Since each client virtually updates, we can directly use the global average as the output. Repeat this training process for a total of $T$ communication rounds to output the final global average model.

Because of the central storage and management of the dual variables on the global server, it significantly reduces storage requirements for lightweight-edged devices, i.e., mobile phones. For the unparticipated clients, we use their unbiased estimations $\mathbb{E}\left[w_i^{t+1} \mid w^t\right] = \mathbb{E}_{\mathcal{P}^t}\left[\frac{1}{P} \sum_{i \in \mathcal{P}^t} w_i^{t+1} \mid w^t\right]$ to construct the virtual dual update, which maintains a continuous update of local dual variables.

For the global averaged dual variable $\overline{\lambda}$, we can reformulate its update as $\overline{\lambda}^{t+1} = \overline{\lambda}^t + \rho(\overline{\theta}^{t+1} - \theta^t)$ which also could be approximated as a virtual all participation case. This efficiently alleviates the *dual drift* between $\theta^t$ and $\lambda_i^t$ and also ensures fast iteration of the global dual variable in the training, which constitutes an efficient federated framework.

## 4 Theoretical Analysis

In this part, we mainly introduce the theoretical analysis of the optimization and generalization efficiency of our proposed *A-FedPD* method. We first introduce the assumptions adopted in our proofs. Optimization analysis is stated in Sec.4.1 and generalization analysis is stated in Sec.4.2.

**Assumption 1 (Smoothness)** *The local function $f_i(\cdot)$ satisfies the $L$-smoothness property, i.e.,* $\|\nabla f_i(\theta_1) - \nabla f_i(\theta_2)\| \le L\|\theta_1 - \theta_2\|$.

**Assumption 2 (Lipschitz continuity)** *For $\forall \theta_1, \theta_2 \in \mathbb{R}^d$, the global function $f(\cdot)$ satisfies the Lipschitz-continuity, i.e.,* $\|f(\theta_1) - f(\theta_2)\| \le G\|\theta_1 - \theta_2\|$.

Optimization analysis only adopts Assumption 1. Generalization analysis adopts both assumptions that were followed from the previous work on analyzing the stability [Hardt et al., 2016, Lei and Ying, 2020, Zhou et al., 2021, Sun et al., 2023e,d,c]. Moreover, we consider that the minimization of each local Lagrangian problem achieves the $\epsilon$-inexact solution during each local training process, i.e. $\|\nabla \mathcal{L}_i\|^2 \le \epsilon$ [Zhang et al., 2021, Li et al., 2023a, Gong et al., 2022, Wang et al., 2022]. This consideration is more aligned with the practical scenarios encountered in the empirical studies for non-convex optimization. In fact, it is precisely because the errors from local inexact solutions can be excessively large that the *dual drift* problem is further exacerbated.

## 4.1 Optimization

In this part, we introduce the convergence analysis of the proposed *A-FedPD* method.

**Theorem 1** *Let non-convex objective $f$ satisfies Assumption 1, let $\rho$ be selected as a non-zero positive constant, $\{\overline{\theta}^t\}_{t=0}^T$ sequence generated by algorithm 1 satisfies:*

$$\frac{1}{T}\sum_{t=1}^{T}\mathbb{E}\|\nabla f(\overline{\theta}^t)\|^2 \leq \frac{\rho\left[f(\overline{\theta}^1) - f^\star\right] + R_0}{T} + \mathcal{O}(\epsilon), \tag{8}$$

*where $f^\star$ is the optimum and $R_0 = \frac{1}{C}\sum_{i\in\mathcal{C}}\mathbb{E}_t\|\theta_i^1 - \theta^0\|^2$ is the first local training volumes.*

**Remark 1.1** *To achieve the $\epsilon$ error, the A-FedPD requires $\mathcal{O}(\epsilon^{-1})$ rounds, yielding $\mathcal{O}(1/T)$ convergence rate. Concretely, the federated primal dual methods can locally train more and communicate less, which empowers it a great potential in the applications. Our analysis is consistent with the previous understandings [Zhang et al., 2021, Durmus et al., 2021, Gong et al., 2022, Li et al., 2023a].*

**Remark 1.2** *Generally, the federated primal-dual methods require a long local interval. Zhang et al. [2021], Gong et al. [2022], Wang et al. [2022] have summarized the corresponding selections of $K$ for different local optimizers. To complete the analysis, we just list a general selection of the local interval $K$. Specifically, to achieve the $\epsilon$ error, local interval $K$ of A-FedPD can be selected as $\mathcal{O}(\epsilon^{-1})$ with total $\mathcal{O}(\epsilon^{-2})$ sample complexity in the training. Due to the page limitation, we state more discussions in the Appendix B.2.2.*

## 4.2 Generalization

In this part, we explore the efficiency of *A-FedPD* from the stability and generalization perspective, which could also be extended to the common *primal dual*-family in the federated learning community. We first introduce the setups and assumptions and then demonstrate the main theorem and discussions.

**Setups.** To understand the stability and generalization efficiency, we follow Hardt et al. [2016], Lei and Ying [2020], Zhou et al. [2021], Sun et al. [2023e] to adopt the uniform stability analysis to measure its error bound. To learn the generalization gap $\mathbb{E}\left[F(\theta^T) - f(\theta^T)\right]$ where $\theta^T$ is generated by a stochastic algorithm, we could study its stability gaps. We consider a joint client set $\mathcal{C}$ (union dataset) for training. Each client $i$ has a private dataset $\mathcal{S}_i$ with total $S$ samples which are sampled from the unknown distribution $\mathcal{D}_i$. To explore the stability gaps, we construct a mirror dataset $\hat{\mathcal{C}}$ that there is at most one different data sample from the raw dataset $\mathcal{C}$. Let $\theta^T$ and $\hat{\theta}^T$ be two models trained on $\mathcal{C}$ and $\hat{\mathcal{C}}$ respectively. Therefore, the generalization of a uniformly stable method satisfies:

$$\mathbb{E}\left[|F(\theta^T) - f(\theta^T)|\right] \leq \sup_{\xi}\mathbb{E}\left[|f(\theta^T,\xi) - f(\hat{\theta}^T,\xi)|\right] \leq \varepsilon. \tag{9}$$

**Key properties.** From the local training, we can first upper bound the local stability. To compare the difference between vanilla *SGD* updates and *primal dual*-family updates, we can reformulate them:

$$\begin{cases} \theta_{i,k+1}^t - \theta^t &= (\theta_{i,k}^t - \theta^t) + \eta^t g_{i,k}^t, \\ \theta_{i,k+1}^t - \theta^t &= (1 - \eta^t\rho)(\theta_{i,k}^t - \theta^t) + \eta^t(g_{i,k}^t + \lambda_i). \end{cases} \tag{10}$$

The above update is for vanilla *FedAvg* and the below update is for *primal dual*-family. When the dual variables are ignored, local update $\theta_{i,k}^t - \theta^t$ in *primal dual* could be considered as a stable decayed sequence with $1 - \eta^t\rho$ that has a constant upper bound. Based on this, we can provide a tighter generalization error bound for the *primal dual*-family methods in FL than the vanilla *FedAvg* method.

**Theorem 2** *Let non-convex objective $f$ satisfies Assumption 1 and 2 and $H = \sup_{\theta,\xi} f(\theta,\xi)$, after $T$ communication rounds training with Algorithm 1, the generalization error bound achieves:*

$$\mathbb{E}\left[F(\theta^T) - f(\theta^T)\right] \leq \frac{\kappa_c}{CS}(HPT)^{\frac{\mu L}{1+\mu L}}, \tag{11}$$

*where $\mu$ is a constant related to the learning rate and $\kappa_c = 4\left(G^2/L\right)^{\frac{1}{1+\mu L}}$ is a constant.*

**Remark 2.1** *We assume that the total number of data samples participating in the training is $CS$ and the total iterations of the training are $KT$. Hardt et al. [2016] prove that on non-convex objectives, vanilla SGD method achieves $\mathcal{O}((TK)^{\frac{\mu L}{1+\mu L}}/CS)$ error bound. Compared with SGD, FL*

Table 2: Test accuracy on the CIFAR-10 / 100 dataset. We fix the total client $C = 100$ and $P = 10$ under training local 50 iterations. We test 3 setups of IID, Dir-1.0, and Dir-0.1 on each dataset. Each group is tested on LeNet (upper portion) and ResNet-18 (lower portion) models. Each results are tested with 4 different random seeds. "−" means can not stably converge. "Family" distinguishes whether the algorithm is a primal method (P) or a primal dual method (PD) and "Local Opt" distinguishes whether the algorithm adopts SGD-based or SAM-based local optimizer.

| | FAMILY | LOCAL OPT | CIFAR-10 | | | CIFAR-100 | | |
| --- | --- | --- | --- | --- | --- | --- | --- | --- |
| | | | IID | DIR-1.0 | DIR-0.1 | IID | DIR-1.0 | DIR-0.1 |
| FEDAVG | P | SGD | $81.87_{\pm.12}$ | $80.58_{\pm.15}$ | $75.57_{\pm.27}$ | $40.11_{\pm.17}$ | $39.65_{\pm.07}$ | $38.37_{\pm.14}$ |
| FEDCM | P | SGD | $80.34_{\pm.14}$ | $79.31_{\pm.33}$ | $72.89_{\pm.37}$ | $43.33_{\pm.13}$ | $42.35_{\pm.25}$ | $37.11_{\pm.51}$ |
| SCAFFOLD | P | SGD | $84.25_{\pm.16}$ | $83.61_{\pm.14}$ | $78.66_{\pm.29}$ | $49.65_{\pm.06}$ | $49.11_{\pm.14}$ | $46.36_{\pm.30}$ |
| FEDSAM | P | SAM | $83.22_{\pm.09}$ | $81.94_{\pm.13}$ | $77.41_{\pm.36}$ | $43.02_{\pm.09}$ | $42.83_{\pm.29}$ | $42.29_{\pm.23}$ |
| FEDDYN | PD | SGD | $84.49_{\pm.22}$ | $84.20_{\pm.14}$ | $79.51_{\pm.13}$ | $50.27_{\pm.11}$ | $49.64_{\pm.21}$ | $46.30_{\pm.26}$ |
| FEDSPEED | PD | SAM | $86.01_{\pm.18}$ | $85.11_{\pm.21}$ | $80.86_{\pm.18}$ | $54.01_{\pm.15}$ | $53.45_{\pm.23}$ | $51.28_{\pm.18}$ |
| A-FEDPD | PD | SGD | $85.31_{\pm.14}$ | $84.94_{\pm.13}$ | $80.28_{\pm.20}$ | $51.41_{\pm.15}$ | $51.17_{\pm.17}$ | $48.15_{\pm.28}$ |
| A-FEDPDSAM | PD | SAM | $\mathbf{86.47}_{\pm.18}$ | $\mathbf{85.90}_{\pm.29}$ | $\mathbf{81.96}_{\pm.19}$ | $\mathbf{55.56}_{\pm.27}$ | $\mathbf{54.62}_{\pm.16}$ | $\mathbf{53.15}_{\pm.19}$ |
| FEDAVG | P | SGD | $81.67_{\pm.21}$ | $80.94_{\pm.17}$ | $76.24_{\pm.35}$ | $44.68_{\pm.21}$ | $44.27_{\pm.25}$ | $41.64_{\pm.27}$ |
| FEDCM | P | SGD | $84.22_{\pm.17}$ | $82.85_{\pm.21}$ | $76.93_{\pm.32}$ | $50.04_{\pm.16}$ | $48.66_{\pm.28}$ | $44.07_{\pm.30}$ |
| SCAFFOLD | P | SGD | $84.31_{\pm.14}$ | $83.70_{\pm.11}$ | $78.70_{\pm.21}$ | $50.69_{\pm.21}$ | $50.28_{\pm.21}$ | $47.12_{\pm.34}$ |
| FEDSAM | P | SAM | $83.79_{\pm.28}$ | $82.58_{\pm.19}$ | $77.83_{\pm.27}$ | $48.66_{\pm.29}$ | $48.42_{\pm.19}$ | $45.03_{\pm.22}$ |
| FEDDYN | PD | SGD | $83.71_{\pm.26}$ | $82.66_{\pm.15}$ | $79.44_{\pm.25}$ | — | — | — |
| FEDSPEED | PD | SAM | $86.90_{\pm.18}$ | $85.92_{\pm.24}$ | $81.47_{\pm.19}$ | $53.22_{\pm.28}$ | $52.75_{\pm.16}$ | $49.66_{\pm.13}$ |
| A-FEDPD | PD | SGD | $85.11_{\pm.12}$ | $84.33_{\pm.16}$ | $81.05_{\pm.28}$ | $48.15_{\pm.22}$ | $48.02_{\pm.29}$ | $46.24_{\pm.26}$ |
| A-FEDPDSAM | PD | SAM | $\mathbf{87.44}_{\pm.13}$ | $\mathbf{86.46}_{\pm.25}$ | $\mathbf{82.48}_{\pm.21}$ | $\mathbf{55.30}_{\pm.23}$ | $\mathbf{53.49}_{\pm.17}$ | $\mathbf{50.31}_{\pm.23}$ |

*adopts the cyclical local training and partial participation mechanism which further increases the stability error. Sun et al. [2023d] learn a fast rate on sample size as $\mathcal{O}((PKT)^{\frac{\mu L}{1+\mu L}}/CS)$ under the Lipschitz assumption only. However, primal dual-family can achieve faster rate $\mathcal{O}((PT)^{\frac{\mu L}{1+\mu L}}/CS)$ in FL, which is due to the stable iterations in Eq.(10). It guarantees that the local training could be bounded in a constant order even under the fixed learning rate. From the local training perspective, primal dual-family in FL can support a very long local interval K without losing stability. This property is also proven in its optimization progress, that the primal dual-family could adopt a larger local interval to accelerate the training and reduce the communication rounds. In general training, especially in situations where communication bandwidth is limited and frequent communication is not possible, the primal dual-family in FL could achieve a more stable result than the general methods. Our analysis further confirms its good adaptivity in FL. Due to page limitation, we summarize some recent results of the generalization error bound in Appendix B.3.2.*

## 5   Experiments

In this section, we introduce the experiments conducted to validate the efficiency of our proposed *A-FedPD* and a variant *A-FedPDSAM* (see details in Appendix A.1). We first introduce experimental setups and benchmarks, and then we show the empirical studies.

**Backbones and Datasets.** In our experiments, we adopt LeNet LeCun et al. [1998] and ResNet He et al. [2016] as backbones. We follow previous work to test the performance of benchmarks on the CIFAR-10 / 100 dataset Krizhevsky et al. [2009]. We introduce the heterogeneity to split the raw dataset to local clients with independent Dirichlet distribution Hsu et al. [2019] controlled by a concentration parameter. In our setups, we mainly test the performance of the IID, Dir-1.0, and Dir-0.1 splitting. The Dir-1.0 represents the low heterogeneity and Dir-0.1 represents the high heterogeneity. We also adopt the sampling with replacement to further enhance the heterogeneity.

**Setups.** We test the accuracy experiments on $C = 100$ and $P/C = 10\%$, which is also the most popular setup in the FL community. In the comparison experiments, we test the participated ratio $P/C = [5\%, 10\%, 20\%, 50\%, 80\%, 100\%]$ and local interval $K = [10, 20, 50, 100, 200]$ respectively. In each setup, for a fair comparison, we freeze the most of hyperparameters for all methods. We fix total communication rounds $T = 800$ except for the ablation studies.

**Baselines.** *FedAvg* [McMahan et al., 2017] is the fundamental paradigm in FL scenarios. *FedCM* [Xu et al., 2021], *SCAFFOLD* [Karimireddy et al., 2020] and *FedSAM* [Qu et al., 2022] are three classical SOTA methods in the federated primal average family. *FedDyn* [Durmus et al., 2021] and

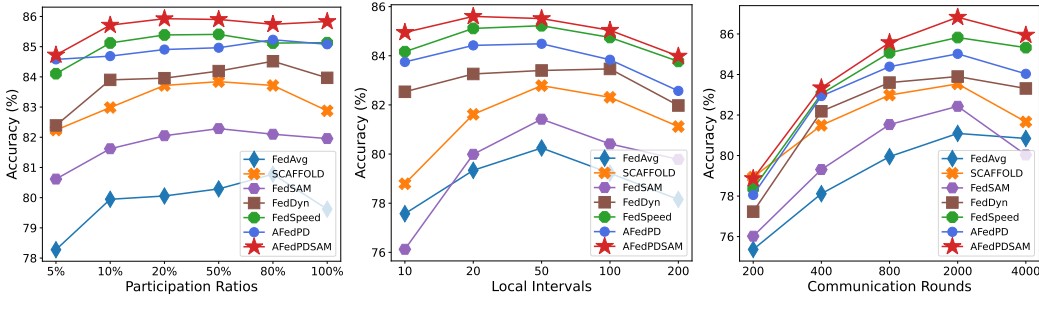

| (a) Different Participation Ratios. | (b) Different Local Intervals. | (c) Different Rounds. |

Figure 2: Test of the proposed *A-FedPD* method on setups of different participation ratios, different local intervals, and different rounds. In these experiments, we fix the total training data samples and total training iterations and then learn their variation trends.

*FedSpeed* Sun et al. [2023b] are relatively stable federated primal dual methods. A more detailed introduction of these methods is presented in Table 2, including the family-basis and local optimizer.

## 5.1 Experiments

In this part, we introduce the phenomena observed in our empirical studies. We primarily investigated performance comparisons, including settings with different participation rates, various local intervals, and different numbers of communication rounds. Then we report the comparison of wall-clock time.

**Performance Comparison.** Table 2 shows the test accuracy on CIFAR-10 / 100 dataset. The vanilla *FedAvg* provides the standard bars as the baseline. In the *federated primal average* methods, The *FedCM* is not stable enough and is largely affected by different backbones, which is caused by the forced consistency momentum and may introduce very large biases. *SCAFFOLD* and *FedSAM* performs well. However, both are less than the *primal dual*-based methods. In summary, *SCAFFOLD* is on average 3.2% lower than *FedSpeed*. As heterogeneity increases, *SCAFFOLD* drops on average about 5.6% on CIFAR-10 and 3.43% on CIFAR-100. In the *federated primal dual* methods, we can clearly see that *FedDyn* is not stable in the training. It performs well on the LeNet, which could achieve at least 1% improvements than *SCAFFOLD*. However, when the task becomes difficult, i.e., ResNet-18 on CIFAR-100, its accuracy is affected by the "dual drift" and drops quickly. To maintain stability, we have to select some weak coefficients to stabilize it and finally get a lower accuracy. Our proposed *A-FedPD* could efficiently alleviate the negative impacts of the "dual drift". It performs about on average 0.8% higher than *FedDyn* on the CIFAR-10 dataset. When *FedDyn* has to compromise the hyperparameters and becomes extremely unstable on ResNet-18 on the CIFAR-100 dataset, *A-FedPD* still performs stably. It's also very scalable. When we introduce the *SAM* optimizer to replace the vanilla *SGD*, it could achieve higher performance.

**Different Participation Ratios.** In this part we compare the sensitivity to the participation ratios. In each setup, we fix the scale as 100 and the local interval as 50 iterations. Active ratio is selected from $[5\%, 10\%, 20\%, 50\%, 80\%, 100\%]$ as shown in Figure 2 (a). Under frozen hyperparameters, all methods perform well on each selection. The best performance is approximately located in the range of $[20\%, 80\%]$. Our proposed methods achieve high efficiency in handling large-scale training, which performs more steadily than other benchmarks across all selections.

**Different Local Intervals.** In this part we compare the sensitivity to the local intervals. In each setup, we fix the scale as 100 and the participation as 10%. Local interval $K$ is selected from $[10, 20, 50, 100, 200]$ as shown in Figure 2 (b). More local training iterations usually mean more overfitting on the local dataset, which leads to a serious "client drift" issue. All methods will be affected by this and drop accuracy. It is a trade-off in selecting the local interval $K$ to balance both optimization efficiency and generalization stability. Our proposed methods still could achieve the best performance even on the very long local training iterations.

**Different Communication Rounds.** In this part, we compare the sensitivity to the communication rounds. In each setup, we fix total iterations $TK = 40,000$. Communication round $T$ is selected from $[200, 400, 800, 2000, 4000]$ as shown in Figure 2 (c). We always expect the local clients can handle more and communicate less, which will significantly reduce the communication costs. In the

experiments, our proposed methods could achieve higher efficiency than the benchmarks. *A-FedPD* saves about half the communication overhead compared to *SCAFFOLD*, and about one-third of *FedDyn*. Under favorable communication bandwidths, they can achieve SOTA performance.

**Wall-clock Time Efficiency.** In this part we test the practical wall-clock time comparisons as shown in Figure 3. Though some methods are communication-efficiency, complicated calculations hinder the real efficiency in wall-clock training time. Though *FedSpeed* and *AFedPDSAM* perform well at the end, additional calculations per single round make their early-stage competitiveness lower. *AFedPD* and *SCAFFOLD* consume fewer time costs, hence achieving better results at the early stage. Without considering training time costs, *AFedPDSAM* achieves the SOTA results at the end. Detailed comparisons are stated in Sec.A.4.4.

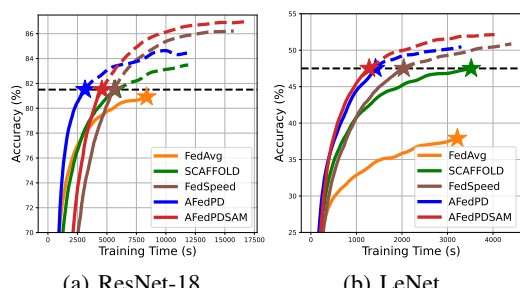

(a) ResNet-18.  (b) LeNet.

Figure 3: Wall-clock time test of training process after total of 600 communication rounds.

## 6 Conclusion

In this paper, we first review the development of the *federated primal dual* methods. Under the exploration of the experiments, we point out a serious challenge that hinders the efficiency of such methods, which is summarized as the "dual drift" problem due to the mismatched primal and dual variables in the partial participation manners. Furthermore, we propose a novel *A-FedPD* method to alleviate this issue via constructing virtual dual updates for those unparticipated clients. We also theoretically learn its convergence rate and generalization error bound to demonstrate its efficiency. Extensive experiments are conducted to validate its significant performance.

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

In this part, we introduce the appendix. We introduce the additional experiments in Sec.A including backgrounds, setups, hyperparameters selections, and some figures of experiments. We introduce the theoretical proofs in Sec.B.

**Limitations.** To avoid the dual drifts, we propose the virtual dual updates to align the old dual variables with the new global model. This requires those dual variables of non-active clients to be updated on the global server, yielding more storage costs. As a trade-off, our method applies additional variable assistance to greatly improve the stability of such algorithms. It also is an interesting future study to approximate the virtual dual update on the local clients.

# A  Additional Experiments

## A.1  Benchmarks

We select 6 classical state-of-the-art (SOTA) benchmarks as baselines in our paper, including (a) *primal*: *FedAvg* McMahan et al. [2017], *SCAFFOLD* Karimireddy et al. [2020], *FedCM* Xu et al. [2021], *FedSAM* Qu et al. [2022]; (b) *primal dual*: *FedADMM* Wang et al. [2022], *FedDyn* Wang et al. [2022], *FedSpeed* Sun et al. [2023b]. We mainly focus on infrastructure improvements instead of combinations of techniques. In the *primal* group, *SCAFFOLD* and *FedCM* alleviate "client drift" via variance reduction and client-level momentum correction respectively. *FedSAM* introduce the *Sharpeness Aware Minimization* (SAM) Foret et al. [2020] in vanilla *FedAvg* to smoothen the loss landscape. In the *primal dual* group, we select the *FedDyn* as the stable basis under partial participation. We also show the instability of the vanilla *FedADMM* to show the negative impacts of the "*dual drift*". *FedSpeed* introduces the *SAM* in vanilla *FedADMM* / *FedDyn*. We also test the variant of *SAM* version of our proposed *A-FedPD* method, which is named *A-FedPDSAM*.

---

**Algorithm 2** A-FedPDSAM Algorithm

---

**Input:** $\theta^0, \theta_i^0, T, K, \lambda_i^0, \rho$
**Output:** global average model
1: **Initialization** : $\theta_i^0 = \theta^0, \lambda_i^0 = 0$.
2: **for** $t = 0, 1, 2, \cdots, T - 1$ **do**
3:  randomly select active clients set $\mathcal{P}^t$ from $\mathcal{C}$
4:  **for** client $i \in \mathcal{P}^t$ **in parallel do**
5:   receive $\lambda_i^t, \theta^t$ from the global server
6:   $\theta_i^{t+1} = LocalTrain(\lambda_i^t, \theta^t, \eta^t, K)$
7:   send $\theta_i^{t+1}$ to the global server
8:  **end for**
9:  $\overline{\theta}^{t+1} = \frac{1}{P} \sum_{i \in \mathcal{P}^t} \theta_i^{t+1}$
10:  $\lambda_i^{t+1} = D\text{-}Update(\lambda_i^t, \theta^t, \theta_i^{t+1}, \overline{\theta}^{t+1}, \mathcal{P}^t)$
11:  $\overline{\lambda}^{t+1} = \frac{1}{C} \sum_{i \in \mathcal{C}} \lambda_i^{t+1}$
12:  $\theta^{t+1} = \overline{\theta}^{t+1} + \frac{1}{\rho} \overline{\lambda}^{t+1}$
13: **end for**
14: return global average model

$\Diamond$ *LocalTrain*: (Optimize Eq.(4))

**Input:** $\lambda_i^t, \theta^t, \eta^t, K$
**Output:** $\theta_{i,K}^t$
1: **for** $k = 0, 1, 2, \cdots, K - 1$ **do**
2:  select a minibatch $\mathcal{B}$
3:  $g_{i,k}^t = \nabla f_i(\theta_{i,k,t} + \rho \frac{\nabla f_i(\theta_{i,k,t}, \mathcal{B})}{\|\nabla f_i(\theta_{i,k,t}, \mathcal{B})\|}, \mathcal{B})$
4:  $\theta_{i,k+1}^t = \theta_{i,k}^t - \eta^t(g_{i,k}^t + \lambda_i^t + \rho(\theta_{i,k}^t - \theta^t))$
5: **end for**
$\Diamond$ *D-Update*: (update dual variables)

**Input:** $\lambda_i^t, \theta^t, \theta_i^{t+1}, \overline{\theta}^{t+1}, \mathcal{P}^t$
**Output:** $\lambda_i^{t+1}$
1: **if** $i \in \mathcal{P}^t$ **then**
2:  $\lambda_i^{t+1} = \lambda_i^t + \rho_t(\theta_i^{t+1} - \theta^t)$
3: **else**
4:  $\lambda_i^{t+1} = \lambda_i^t + \rho_t(\overline{\theta}^{t+1} - \theta^t)$
5: **end if**

---

## A.2  Hyperparameters Selection

We first introduce the hyperparameter selections. To fairly compare the efficiency of the benchmarks, we fix the most of hyperparameters, including the initial global learning rate, the initial learning rate, the weight decay coefficient, and the local batchsize. The other hyperparameters are selected properly on a grid search within the valid range. The specific hyperparameters of specific methods are defined in the experiments. We report the corresponding selections of their best performance, which is summarized in the following Table 3.

The global learning rate is fixed in our experiments. Though Asad et al. [2020] propose to adopt the double learning rate decay both on the global server and local client can make training more efficient, we find some methods will easily over-fit under a global learning rate decay. For the weight decay coefficient, we recommend to adopt 0.001. Actually, we find that adjusting it still can improve the performance of some specific methods. One of the most important hyperparameters is learning rate

Table 3: Hyperparameters selections of benchmarks.

|  | Grid Search | FedAvg | FedCM | SCAFFOLD | FedSAM | FedDyn | FedSpeed |
|---|---|---|---|---|---|---|---|
| global learning rate | [0.1, 1.0] | 1.0 | | | | | |
| local learning rate | [0.01, 0.1, 1.0] | 0.1 | | | | | |
| weight decay | [0.0001, 0.001, 0.01] | 0.001 | | | | | |
| learning rate decay | [0.995, 0.998, 0.9998, 1] | 0.998 | 0.998 | 0.998 | 0.998 | 0.998 / 1 | 0.998 / 1 |
| batchsize | [10, 20, 50, 100] | 50 | 50 | 50 | 50 | 20 / 50 | 50 |
| client-level momentum | [0.05, 0.1, 0.2, 0.5] | | 0.1 | | | | |
| proxy coefficient | [0.001, 0.01, 0.1, 1.0] | | | | | 0.1 / 0.001 | 0.1 |
| SAM perturbation | [0.01, 0.05, 0.1, 0.5] | | | | 0.05 | | 0.1 |
| SAM eps | [1e-2, 1e-5, 1e-8] | | | | 1e-2 | | all |

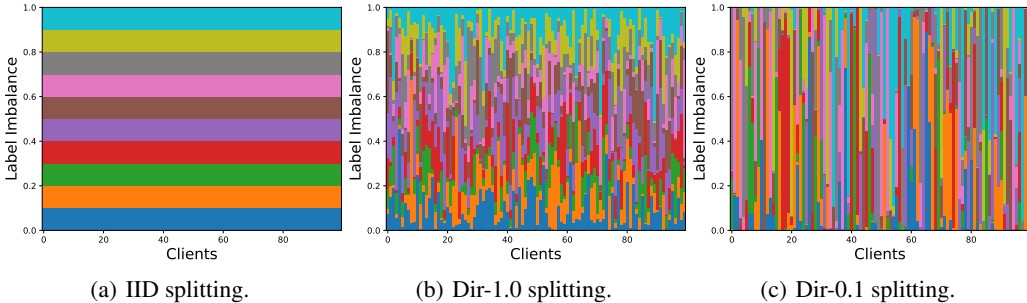

(a) IID splitting.    (b) Dir-1.0 splitting.    (c) Dir-0.1 splitting.

Figure 4: Label ratios under different splitting manners. Different color means the samples are in different labels. We show the different splitting distributions on a total of 100 clients.

decay. Generally, we use $d^T = 1 / 0.8 / 0.1 / 0.005$ to select the proper $d$ as a decayed coefficient, which means the level of the initial learning rate will be decayed after $T$ communication rounds. We follow previous studies to fix the learning rate within the communication round. Another important hyperparameter is the batchsize. In our experiments, we fixed the local interval which means the fixed local iterations. Due to the sample size being fixed, different batchsizes mean different training epochs. Generally speaking, training epochs always decide the optimization level on the local optimum. A too-long interval always leads to overfitting to the local dataset and falling into the serious "client drift" problem [Karimireddy et al., 2020]. In our experiments, we control it to stop when the local client is optimized well enough, i.e., a proper loss or accuracy. For the specific hyperparameters for each method, we directly grid search from the selections. For *A-FedPDSAM* method, the selections are consistent with the *FedDyn* and *FedSpeed* methods except for the learning rate decay is fixed as 0.998. To alleviate the "dual drift", we properly reduce the proxy coefficient for the *FedDyn* on those difficult tasks to maintain stable training.

## A.3 Dataset and Splitting

We use the CIFAR-10 / 100 datasets to validate the efficiency, which is widely used to verify the federated efficiency [McMahan et al., 2017, Karimireddy et al., 2020, Li et al., 2020, Xu et al., 2021, Durmus et al., 2021, Gong et al., 2022, Wang et al., 2022, Fan et al., 2022, Caldarola et al., 2022, Sun et al., 2023b,a, Li et al., 2023a, Fan et al., 2024a,b]. The total dataset of both contain 50,000 training samples and 10,000 test samples of 10 / 100 classes. Each is a colorful image in the size of 32×32. We follow the training as the vanilla SGD to add data augmentation without additional operations.

**Label Heterogeneity.** For the dataset splitting, we adopt the label imbalanced splitting under the Dirichlet manners. We first generate a distribution matrix and then generate a random number to sample each data. To further enhance the local heterogeneity, we also adopt the sampling with replacement, which means one data sample may exist on several clients simultaneously. This is more related to real-world scenarios because of the local unknown dataset distribution. We generate the matrices in Fig. 4 to show their distribution differences. We can clearly see that Dir-0.1 introduces a very large heterogeneity in that there is almost one dominant class in a client. Dir-1.0 handles approximately 3 classes in one client. Actually, in practical scenarios, label imbalance may be the most popular heterogeneity because we often expect both the local task and local dataset to be still unknown to others. For instance, client $i$ may be an expert on task $A$, and client $j$ may be an expert

on another task $B$. To combine the tasks $A$ and $B$, if we can directly merge them with a training policy without beforehand knowing the tasks, then it must further enhance local privacy.

**Brightness Heterogeneity.** To further simulate the real-world manners, we allow different clients to change the brightness and ratios of different color channels. This corresponds to different sources of data collected by different local clients. We show some samples in Fig. 5 to show how different they are on different clients. Specifically, after splitting the local dataset, we will calculate the average brightness of each local dataset. Then we generate a noise from Gaussian to randomly change the brightness and one of the color channels, which means that even similar samples have large color differences on different clients.

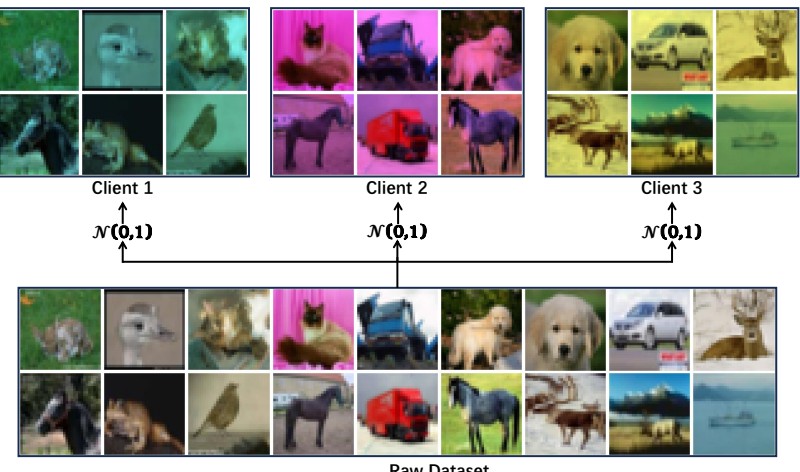

Figure 5: Introducing the brightness biases to different clients. We calculate the average brightness to control each sample to a proper state. Each client will randomly sample a Gaussian noise to perturb the local samples.

## A.4 Additional Experiments

### A.4.1 Some Training Curves

In Figure 6 we can see some experiment curves. From the (a), (b), (c), and (d), we can clearly see that the *A-FedPD* method achieves the fastest convergence rate on each setup. Due to the virtual update on the dual variables, we can treat those unparticipated clients as virtually trained ones. This empowers the *A-FedPD* method with a great convergence speed. Especially on the IID dataset, due to the local datasets being similar (drawn from a global distribution), the expectation of the updated averaged models is the same as that of the updated local model with lower variance. Then we could approximate the local dual update as the global one. This greatly speeds up the training time. We also can see the fast rate of the *FedDyn* method. However, due to its lagging dual update, it will be slower than the *A-FedPD* method. As for the *SAM* variant, it introduces an additional perturbation step that could avoid overfitting. Therefore, its loss does not drop quickly because of the additional ascent step.

From the (e), (f), (g), and (h), we can clearly see the improvements of *A-FedPD* and *A-FedPDSAM* methods. From the basic version, *A-FedPD* could achieve higher performance due to the virtual dual updates. After incorporating *SAM*, local clients could efficiently alleviate overfitting. The global model becomes more stable and could achieve the SOTA results. We will learn the consistency performance in the next part.

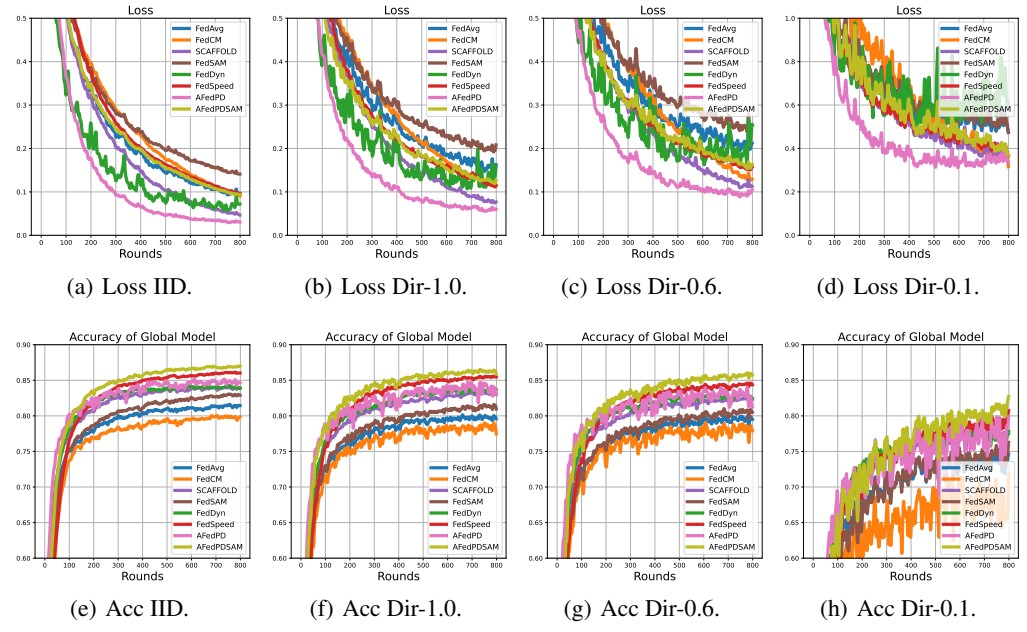

Figure 6: Loss and accuracy curves in the experiments.

### A.4.2 Primal Residual and Dual Residual

In the *primal dual* methods, due to the joint solution on both the primal and dual problems, it leads to an issue that both residuals should maintain a proper ratio. Therefore, the quantity of global updates in the training could be considered as a residual for the dual feasibility condition. The same, the constraint itself could be considered as the primal residual. Then we consider Eq.(3) objectives. The constraint is the global consensus level $\theta - \theta_i$ and the global update is $\theta^{t+1} - \theta^t$. To generally express them, we define the primal residual $p_r^t = \frac{1}{C} \sum_i \|\theta^t - \theta_i^t\|$ and the dual residual $d_r^t = \rho \|\theta^t - \theta^{t-1}\|$. Actually, the primal residual could be considered as the consistency, and the dual residual could be considered as the update. In the training, if we focus more on the dual residual, it leads to a fast convergence on an extremely biased objective that is far away from the true optimal. If we focus more on the primal residual, the local training cannot perform normally for its strong regularizations. Therefore, we must maintain stable trends on both $p_r$ and $d_r$ to implement stable training. In this part, we study the relationships between primal and dual residuals.

As shown in Figure 7, we can clearly see the lower stable ratio between the primal and dual residuals on the *A-FedPD* and *A-FedPDSAM* methods, which indicates that both the primal training and dual training are performed well simultaneously. However, the *federated primal average*-based methods, i.e., *FedAvg* and *SCAFFOLD*, focus more on the primal training which leads to the dual residuals are too small (dual residuals measure the global update; primal residuals measure the global consistency).

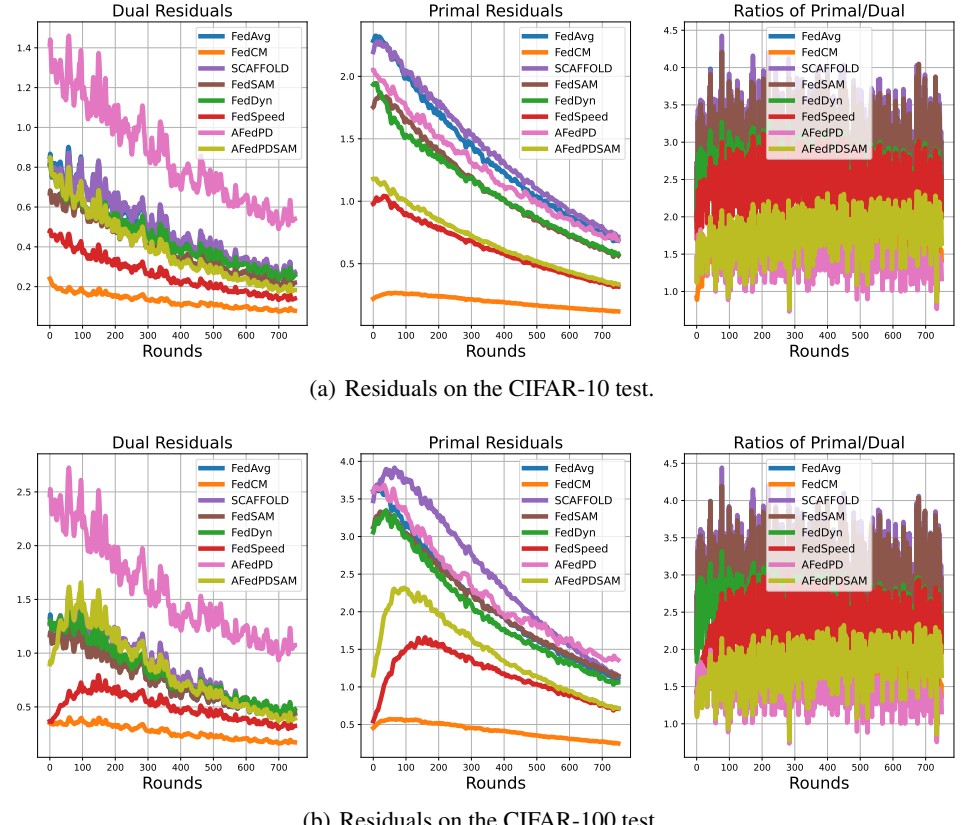

(a) Residuals on the CIFAR-10 test.

(b) Residuals on the CIFAR-100 test.

Figure 7: Loss and accuracy curves in the experiments.

### A.4.3 Communication Efficiency

In this part, we learn the general communication efficiency. We fix all local intervals, clients, participation ratios, and hyperparameters for fairness. We select the different targets as the objective to calculate the communication efficiency.

Table 4: Communication rounds required to achieve the target accuracy.

| CIFAR-10 | FedAvg | SCAFFOLD | FedSAM | FedDyn | FedSpeed | A-FedPD | A-FedPDSAM |
|---|---|---|---|---|---|---|---|
| 81% | 501 | 207 | 345 | 156 | 170 | 131 | 156 |
| | 1× | 2.42× | 1.45× | 3.21× | 2.94× | 3.82× | 3.21× |
| 83.5% | - | 468 | - | 355 | 268 | 252 | 218 |
| | - | 1× | - | 1.31× | 1.74× | 1.85× | 2.14× |
| CIFAR-100 | FedAvg | SCAFFOLD | FedSAM | FedDyn | FedSpeed | A-FedPD | A-FedPDSAM |
| 40% | 772 | 162 | 572 | 173 | 222 | 123 | 126 |
| | 1× | 4.76× | 1.34× | 4.46× | 3.47× | 6.27× | 6.12× |
| 49% | - | 677 | - | 495 | 421 | 303 | 220 |
| | - | 1× | - | 1.36× | 1.61× | 2.23× | 3.07× |

Table 4 shows the communication efficiency among different methods on the CIFAR-10 / 100 dataset trained with LeNet. We calculate the first communication round index of achieving the target accuracy

for comparison. We can clearly see that the communication rounds required for training are saved a lot on the proposed *A-FedPD* and *A-FedPDSAM* methods. It generally accelerates the training process by at least $3\times$ on CIFAR-10 and $6\times$ on CIFAR-100 than the vanilla *FedAvg* method. Compared with the other benchmarks, our proposed method performs stably and efficiently.

### A.4.4 Wall clock Time for Training Costs

Then we further study the wall clock time required in the training. We provide the experimental setups as follows.

Platform: Pytorch 2.0.1        Cuda: 11.7        Hardware: NVIDIA GeForce RTX 2080 Ti

Table 5: Wall clock time required to train 1 round (100 iterations) on **ResNet**.

|  | FedAvg | SCAFFOLD | FedSAM | FedDyn | FedSpeed | A-FedPD | A-FedPDSAM |
|---|---|---|---|---|---|---|---|
| s / round | 22.06 | 36.01 | 39.21 | 34.03 | 56.10 | 37.61 | 60.63 |
|  | $1\times$ | $1.63\times$ | $1.77\times$ | $1.54\times$ | $2.54\times$ | $1.70\times$ | $2.74\times$ |

Table 6: Wall clock time required to train 1 round (100 iterations) on **LeNet**.

|  | FedAvg | SCAFFOLD | FedSAM | FedDyn | FedSpeed | A-FedPD | A-FedPDSAM |
|---|---|---|---|---|---|---|---|
| s / round | 9.82 | 11.42 | 12.53 | 11.76 | 13.61 | 11.71 | 13.90 |
|  | $1\times$ | $1.16\times$ | $1.27\times$ | $1.19\times$ | $1.38\times$ | $1.19\times$ | $1.41\times$ |

Actually, the LeNet is too small for the GPU and the training time does not achieve the capacity, which leads to the close time costs in Table 6. We recommend referring to the cost ratio on the ResNet (Table 5), which is much closer to the real algorithmic efficiency.

# B Proofs.

In this part, we mainly show the proof details of the main theorems in this paper. We will reiterate the background details in Sec.B.1. Then we introduce the important lemmas used in the proof in Sec.B.3.1, and show the proof details of the main theorems in Sec.B.3.2.

## B.1 Preliminaries

Here we reiterate the background details in the proofs. To understand the stability efficiency, we follow Hardt et al. [2016], Lei and Ying [2020], Zhou et al. [2021], Sun et al. [2023e] to adopt the uniform stability analysis in our analysis. Through refining the local subproblem of the solution of the Augmented Lagrangian objective, we provide the error term of the primal and dual terms and their corresponding complexity bound in FL.

Before introducing the important lemmas, we re-summarize the pipelines in the analysis. According to the federated setups, we assume a global server coordinates a set of local clients $\mathcal{C} \triangleq \{i\}_{i=1}^{C}$ to train one model. Each client has a private dataset $\mathcal{S}_i = \{\zeta_{ij}\}_{j=1}^{S}$. We assume that the global joint dataset is the union of $\{\mathcal{S}_1 \cup \mathcal{S}_2 \cup \cdots \cup \mathcal{S}_C\}$. To study its stability, we assume there is another global joint dataset that contains at most one different data sample from $\mathcal{C}$. Let the index of the different pair be $(i^\star, j^\star)$. We train two global models $\theta^T$ and $\hat{\theta}^T$ on these two global joint datasets respectively and gauge their gaps during the training process.

Then we rethink the local solution of the local augmented Lagrangian function. According to the basic algorithm, we have:

$$\mathcal{L}_i(\theta, \lambda_i^t, \theta^t) = f_i(\theta) + \langle \lambda_i^t, \theta - \theta^t \rangle + \frac{\rho}{2} \|\theta - \theta^t\|^2. \tag{12}$$

To upper bound its stability without loss of generality, we consider adopting the general SGD optimizer to solve the sub-problem via total $K$ iterations with the local learning rate $\eta^t$:

$$\theta_{i,k+1}^t = \theta_{i,k}^t - \eta^t \nabla \mathcal{L}_i(\theta_{i,k}^t, \lambda_i^t, \theta^t) = \theta_{i,k}^t - \eta^t \left[ g_{i,k}^t + \lambda_i^t + \rho \left( \theta_{i,k}^t - \theta^t \right) \right], \tag{13}$$

where $k$ is the index of local iterations ($0 \leq k \leq K$).

## B.2 Optimization

### B.2.1 Important Lemmas

In this part, we mainly introduce some important lemmas adopted in the optimization proofs.

Motivated by Durmus et al. [2021], we assume the local client solves the inexact solution of each local Lagrangian function, therefore we have $\nabla f_i(\theta_i^{t+1}) + \lambda_i^t + \rho(\theta_i^{t+1} - \theta^t) = e$, where $e$ can be considered as an error variable with $\|e\|^2 \leq \epsilon$. This can characterize the different solutions of the local sub-problems. We always expect the error to achieve zero. Durmus et al. [2021] only assume that the local solution is exact and this may be not possible in practice.

**Lemma 1 ([Durmus et al., 2021])** *The conditionally expected gaps between the current averaged local parameters and last averaged local parameters satisfy:*

$$\mathbb{E}_t \|\overline{\theta}^{t+1} - \overline{\theta}^t\|^2 \leq \frac{1}{C} \sum_{i \in \mathcal{C}} \mathbb{E}_t \|\theta_i^{t+1} - \overline{\theta}^t\|^2.$$

**Proof.** *According to the averaged randomly sampling, we have:*

$$\mathbb{E}_t \|\overline{\theta}^{t+1} - \overline{\theta}^t\|^2 = \mathbb{E}_t \|\frac{1}{P} \sum_{i \in \mathcal{P}^t} \theta_i^{t+1} - \overline{\theta}^t\|^2 \leq \frac{1}{P} \mathbb{E}_t \sum_{i \in \mathcal{P}^t} \|\theta_i^{t+1} - \overline{\theta}^t\|^2$$

$$= \frac{1}{P} \mathbb{E}_t \sum_{i \in \mathcal{C}} \|\theta_i^{t+1} - \overline{\theta}^t\|^2 \cdot \mathbb{I}_i \leq \frac{1}{C} \sum_{i \in \mathcal{C}} \mathbb{E}_t \|\theta_i^{t+1} - \overline{\theta}^t\|^2.$$

$\mathbb{I}_i$ *is the indicator function as* $\mathbb{I}_i = 1$ *if* $i \in \mathcal{P}^t$ *else 0.*

**Lemma 2** *Under Assumption 1 and let the local solution be an $\epsilon$-inexact solution, the conditionally expected averaged local updates satisfy:*

$$\left(1 - \frac{4L^2}{\rho^2}\right)\frac{1}{C}\sum_{i\in\mathcal{C}}\mathbb{E}_t\|\theta_i^{t+1} - \overline{\theta}^t\|^2 \le \frac{8L^2}{\rho^2}\frac{1}{C}\sum_{i\in\mathcal{C}}\mathbb{E}_t\|\theta_i^t - \overline{\theta}^t\|^2 + \frac{4L^2}{\rho^2}\mathbb{E}_t\|\nabla f(\overline{\theta}^t)\|^2 + \frac{4\epsilon}{\rho^2}. \quad (14)$$

**Proof.** *First, we reconstruct the update of the dual variable. From the updated rules, we have* $\overline{\lambda}_i^{t+1} - \overline{\lambda}_i^t = \rho(\overline{\theta}^{t+1} - \theta^t)$. *From the first order condition of* $\nabla f_i(\theta_i^{t+1}) + \lambda_i^t + \rho(\theta_i^{t+1} - \theta^t) = e$. *By expanding Lemma 11 in [Durmus et al., 2021], then we have:*

$$\frac{1}{C}\sum_{i\in\mathcal{C}}\mathbb{E}_t\|\theta_i^{t+1} - \overline{\theta}^t\|^2$$

$$= \frac{1}{C}\sum_{i\in\mathcal{C}}\mathbb{E}_t\|\theta_i^{t+1} - \theta^t + \frac{1}{\rho}\overline{\lambda}^t\|^2$$

$$\le \frac{4L^2}{\rho^2}\frac{1}{C}\sum_{i\in\mathcal{C}}\left(\mathbb{E}_t\|\theta_i^{t+1} - \overline{\theta}^t\|^2 + 2\mathbb{E}_t\|\theta_i^t - \overline{\theta}^t\|^2 + \mathbb{E}_t\|\nabla f(\overline{\theta}^t)\|^2\right) + \frac{4\epsilon}{\rho^2}.$$

*Therefore, we can reconstruct its relationship as:*

$$\left(1 - \frac{4L^2}{\rho^2}\right)\frac{1}{C}\sum_{i\in\mathcal{C}}\mathbb{E}_t\|\theta_i^{t+1} - \overline{\theta}^t\|^2 \le \frac{8L^2}{\rho^2}\frac{1}{C}\sum_{i\in\mathcal{C}}\mathbb{E}_t\|\theta_i^t - \overline{\theta}^t\|^2 + \frac{4L^2}{\rho^2}\mathbb{E}_t\|\nabla f(\overline{\theta}^t)\|^2 + \frac{4\epsilon}{\rho^2}.$$

*This completes the proofs.*

**Lemma 3** *Under Assumption 1, the conditionally expected averaged local consistency satisfies:*

$$\frac{1}{C}\sum_{i\in\mathcal{C}}\mathbb{E}_t\|\theta_i^{t+1} - \overline{\theta}^{t+1}\|^2 \le \frac{4}{C}\sum_{i\in\mathcal{C}}\mathbb{E}_t\|\theta_i^{t+1} - \overline{\theta}^t\|^2. \quad (15)$$

**Proof.** *According to the update rules, we have:*

$$\frac{1}{C}\sum_{i\in\mathcal{C}}\mathbb{E}_t\|\theta_i^{t+1} - \overline{\theta}^{t+1}\|^2 = \frac{1}{C}\sum_{i\in\mathcal{C}}\mathbb{E}_t\|\theta_i^{t+1} - \overline{\theta}^t + \overline{\theta}^t - \overline{\theta}^{t+1}\|^2$$

$$\le \frac{2}{C}\sum_{i\in\mathcal{C}}\mathbb{E}_t\|\theta_i^{t+1} - \overline{\theta}^t\|^2 + 2\mathbb{E}_t\|\overline{\theta}^t - \overline{\theta}^{t+1}\|^2$$

$$\le \frac{2}{C}\sum_{i\in\mathcal{C}}\mathbb{E}_t\|\theta_i^{t+1} - \overline{\theta}^t\|^2 + \frac{2}{C}\sum_{i\in\mathcal{C}}\mathbb{E}_t\|\theta_i^{t+1} - \overline{\theta}^t\|^2$$

$$\le \frac{4}{C}\sum_{i\in\mathcal{C}}\mathbb{E}_t\|\theta_i^{t+1} - \overline{\theta}^t\|^2.$$

*This completes the proofs.*

### B.2.2 Proofs

According to the smoothness, we take the conditional expectation on round $t$ and expand the global function as:

$$\mathbb{E}_t\left[f(\overline{\theta}^{t+1})\right] - f(\overline{\theta}^t)$$

$$\le \frac{L}{2}\mathbb{E}_t\|\overline{\theta}^{t+1} - \overline{\theta}^t\|^2 + \mathbb{E}_t\langle\nabla f(\overline{\theta}^t), \overline{\theta}^{t+1} - \overline{\theta}^t\rangle$$

$$= \frac{L}{2}\mathbb{E}_t\|\overline{\theta}^{t+1} - \overline{\theta}^t\|^2 + \mathbb{E}_t\langle\nabla f(\overline{\theta}^t), \frac{1}{C}\sum_{i\in\mathcal{C}}\theta_i^{t+1} - \overline{\theta}^t\rangle$$

$$= \frac{L}{2}\mathbb{E}_t\|\overline{\theta}^{t+1} - \overline{\theta}^t\|^2 + \mathbb{E}_t\langle\nabla f(\overline{\theta}^t), \frac{1}{C}\sum_{i\in\mathcal{C}}\left(\theta_i^{t+1} - \theta^t\right) + \theta^t - \overline{\theta}^t\rangle$$

$$= \frac{L}{2}\mathbb{E}_t\|\overline{\theta}^{t+1} - \overline{\theta}^t\|^2 - \mathbb{E}_t\langle\nabla f(\overline{\theta}^t), \frac{1}{C}\sum_{i\in\mathcal{C}}\frac{1}{\rho}\left(\nabla f_i(\theta_i^{t+1}) + \lambda_i^t - e\right) - \frac{1}{\rho}\overline{\lambda}^t\rangle$$

$$
= \frac{L}{2}\mathbb{E}_t\|\overline{\theta}^{t+1} - \overline{\theta}^t\|^2 - \mathbb{E}_t\langle \nabla f(\overline{\theta}^t), \frac{1}{C}\sum_{i\in\mathcal{C}}\frac{1}{\rho}\nabla f_i(\theta_i^{t+1})\rangle
$$

$$
\leq \frac{L}{2}\mathbb{E}_t\|\overline{\theta}^{t+1} - \overline{\theta}^t\|^2 + \frac{1}{2\rho}\mathbb{E}_t\|\nabla f(\overline{\theta}^t) - \frac{1}{C}\sum_{i\in\mathcal{C}}\nabla f_i(\theta_i^{t+1})\|^2 - \frac{1}{2\rho}\mathbb{E}_t\|\nabla f(\overline{\theta}^t)\|^2
$$

$$
\leq \frac{L}{2}\mathbb{E}_t\|\overline{\theta}^{t+1} - \overline{\theta}^t\|^2 + \frac{1}{2\rho}\frac{1}{C}\sum_{i\in\mathcal{C}}\mathbb{E}_t\|\nabla f_i(\overline{\theta}^t) - \nabla f_i(\theta_i^{t+1})\|^2 - \frac{1}{2\rho}\mathbb{E}_t\|\nabla f(\overline{\theta}^t)\|^2
$$

$$
\leq \frac{L}{2}\mathbb{E}_t\|\overline{\theta}^{t+1} - \overline{\theta}^t\|^2 + \frac{L^2}{2\rho}\frac{1}{C}\sum_{i\in\mathcal{C}}\mathbb{E}_t\|\overline{\theta}^t - \theta_i^{t+1}\|^2 - \frac{1}{2\rho}\mathbb{E}_t\|\nabla f(\overline{\theta}^t)\|^2
$$

$$
\leq \frac{L}{2}\left(1 + \frac{L}{\rho}\right)\frac{1}{C}\sum_{i\in\mathcal{C}}\mathbb{E}_t\|\overline{\theta}^t - \theta_i^{t+1}\|^2 - \frac{1}{2\rho}\mathbb{E}_t\|\nabla f(\overline{\theta}^t)\|^2.
$$

To simplify the expression, we define $R_t = \frac{1}{C}\sum_{i\in\mathcal{C}}\mathbb{E}_t\|\theta_i^{t+1} - \overline{\theta}^t\|^2$, $J_t = \frac{1}{C}\sum_{i\in\mathcal{C}}\mathbb{E}_t\|\theta_i^t - \overline{\theta}^t\|^2$. Actually from Lemma 2 and 3, we can reconstruct the relationship as:

$$
\begin{cases}
\mathbb{E}_t\left[f(\overline{\theta}^{t+1})\right] & \leq f(\overline{\theta}^t) + \frac{L}{2}\left(1 + \frac{L}{\rho}\right)R_t - \frac{1}{2\rho}\mathbb{E}_t\|\nabla f(\overline{\theta}^t)\|^2, \\
\left(1 - \frac{4L^2}{\rho^2}\right)R_t & \leq \frac{32L^2}{\rho^2}R_{t-1} + \frac{4L^2}{\rho^2}\mathbb{E}_t\|\nabla f(\overline{\theta}^t)\|^2 + \frac{4\epsilon}{\rho^2},
\end{cases}
$$

Let the second inequality be multiplied by $q$ and add it to the first, we have:

$$
\mathbb{E}_t\left[f(\overline{\theta}^{t+1})\right] + \left[q\left(1 - \frac{4L^2}{\rho^2}\right) - \frac{L}{2}\left(1 + \frac{L}{\rho}\right)\right]R_t
$$

$$
\leq f(\overline{\theta}^t) + q\frac{32L^2}{\rho^2}R_{t-1} - \left(\frac{1}{2\rho} - \frac{4qL^2}{\rho^2}\right)\mathbb{E}_t\|\nabla f(\overline{\theta}^t)\|^2 + \frac{4q\epsilon}{\rho^2}.
$$

Then we discuss the selection of $q$. First, let $\frac{1}{2\rho} - \frac{4qL^2}{\rho^2} > 0$ be positive, which requires $q < \frac{\rho}{8L^2}$. Then we let the following relationship hold:

$$
q\left(1 - \frac{4L^2}{\rho^2}\right) - \frac{L}{2}\left(1 + \frac{L}{\rho}\right) = q\frac{32L^2}{\rho^2},
$$

Thus it requires $2q = \frac{L(\rho^2 + \rho L)}{\rho^2 - 36L^2} < \frac{\rho}{4L^2}$. We can solve this to get the range of the coefficient $\rho$ as:

$$
\rho^2 - 4L^3\rho - 36L^2 - 4L^4 > 0.
$$

Then $\rho > \mathcal{O}(L^3)$ satisfies all the conditions above.

Therefore, let $q = \frac{\rho}{32L^2}$ and then $q\frac{32L^2}{\rho^2} = \frac{1}{\rho}$. By further relaxing the last coefficient we have:

$$
\mathbb{E}_t\left[f(\overline{\theta}^{t+1})\right] + \frac{1}{\rho}R_t \leq f(\overline{\theta}^t) + \frac{1}{\rho}R_{t-1} - \frac{1}{\rho}\mathbb{E}_t\|\nabla f(\overline{\theta}^t)\|^2 + \frac{\epsilon}{8L^2\rho}.
$$

Taking the full expectation and accumulating the above inequality from $t = 0$ to $T-1$, we have:

$$
\frac{1}{T}\sum_{t=1}^{T}\mathbb{E}\|\nabla f(\overline{\theta}^t)\|^2 \leq \frac{f(\overline{\theta}^1) - \mathbb{E}\left[f(\overline{\theta}^{T+1})\right]}{T} + \frac{R_0 - R_T}{\rho T} + \frac{\epsilon}{8L^2\rho}
$$

$$
\leq \frac{\rho\left[f(\overline{\theta}^1) - f^\star\right] + R_0}{T} + \frac{\epsilon}{8L^2\rho}.
$$

In the last inequality, $f^\star$ is the optimum of the function $f$. For the $R_{-1}$ term, we have $R_0 = \frac{1}{C}\sum_{i\in\mathcal{C}}\mathbb{E}_t\|\theta_i^1 - \overline{\theta}^0\|^2 = \frac{1}{C}\sum_{i\in\mathcal{C}}\mathbb{E}_t\|\theta_i^1 - \theta^0\|^2$ for $\theta^0 = \overline{\theta}^0$.

**Discussions of the optimization errors.** We show some classical results of the generalization errors in the following Table 7. Zhang et al. [2021] provides a first optimization analysis for the federated primal-dual methods. However, it requires all clients to participate in the training in each

round (do not support partial participation). Wang et al. [2022], Gong et al. [2022] proposes to adopt partial participation for the federated primal-dual method and adopt a stronger assumption on the local solution. It proves that even though each local solution is different, the total optimization error can be bounded by the average of the trajectory of each local client. However, the initial bias may be affected by the factor $C$ under some special learning rate. Durmus et al. [2021] further provides a variant to calculate the global dual variable. It provides a lower constant for the $D$ term, which is $\frac{C}{P}D$ (faster than $CD$ in FedADMM). Our proposed method A-FedPD updates the virtual dual variables, which could approximate the full-participation training under the partial participation case. Therefore, compared with the result in [Durmus et al., 2021], we further provide a faster constant for the term $D$ ($\frac{C}{P}\times$ faster than FedDyn on the first term when $\rho$ is selected properly). If the initialization bias $D$ dominates the optimization errors, i.e. training from scratch, A-FedPD can greatly improve the training efficiency.

Table 7: Optimization rate of federated smooth non-convex objectives.

| | Assumption | Optimization | reduce *dual-drift*? |
|---|---|---|---|
| Zhang et al. [2021] | smoothness, $\epsilon$-inexact solution | $\mathcal{O}(\frac{D}{T} + \epsilon)$ | $\times$ |
| Hu et al. [2022] | smoothness, $\epsilon_{i,t}$-inexact solution | $\mathcal{O}(\frac{CD}{PT} + \frac{1}{CT}\sum_{i,t}\epsilon_{i,t})$ | $\times$ |
| Durmus et al. [2021] | smoothness, exact solution | $\mathcal{O}(\frac{CD}{PT} + \frac{R_0}{T})$ | $\times$ |
| our | smoothness, $\epsilon$-inexact solution | $\mathcal{O}(\frac{D}{T} + \frac{R_0}{T} + \epsilon)$ | $\checkmark$ |

**Our Improvements.** In [Zhang et al., 2021], it must rely on the full participation. In [Gong et al., 2022, Wang et al., 2022], the impact of the initial bias is $\frac{C}{P}$ times. In [Durmus et al., 2021], it must rely on the local exact solution, which is an extremely ideal condition. Our results can achieve the $\mathcal{O}(\frac{1}{T})$ rate under the general assumptions and support the partial participation case.

### B.3 Generalization

#### B.3.1 Important Lemmas

In this part, we mainly introduce some important lemmas adopted in our proofs. Let $\hat{\cdot}$ be the corresponding variable trained on the dataset $\hat{\mathcal{C}}$, and then we can explore the gaps of corresponding terms. We first consider the stability definition.

**Lemma 4 (Hardt et al. [2016])** *Under Assumption 1 and 2, the model $\theta^T$ and $\hat{\theta}^T$ are generated on the two different datasets $\mathcal{C}$ and $\hat{\mathcal{C}}$ with the same algorithm. We can track the difference between these two sequences. Before we first select the different sample pairs, the difference is always 0. Therefore, we define an event $\zeta$ to measure whether $\theta^T = \hat{\theta}^T$ still holds at $\tau_0$-th round. Let $H = \sup_{\theta,\xi} f(\theta, \xi) < +\infty$, if the algorithm is uniform stable, we can measure its uniform stability by:*

$$\epsilon_G \leq \sup_{\mathcal{C},\hat{\mathcal{C}},\xi} \mathbb{E}\left[f(\theta^T, \xi) - f(\hat{\theta}^T, \xi)\right] \leq G\mathbb{E}\|\theta^T - \hat{\theta}^T\| + \frac{HP\tau_0}{CS}. \tag{16}$$

**Proof.** *By expanding the inequality, we have:*

$$\mathbb{E}\left[|f(\theta^T, \xi) - f(\hat{\theta}^T, \xi)|\right]$$
$$\leq P(\zeta)\mathbb{E}\left[|f(\theta^T, \xi) - f(\hat{\theta}^T, \xi)| \mid \zeta\right] + P(\zeta^c)\mathbb{E}\left[|f(\theta^T, \xi) - f(\hat{\theta}^T, \xi)| \mid \zeta^c\right]$$
$$\leq G\mathbb{E}\left[\|\theta^T - \hat{\theta}^T\| \mid \zeta\right] + HP(\zeta^c).$$

*Here we assume that the difference pairs are selected on $\tau$-th round, therefore we have:*

$$P(\zeta^c) = P(\tau \leq \tau_0) \leq \sum_{t=0}^{\tau_0} P(\tau = t) = \sum_{t=0}^{\tau_0} P(i^\star \in \mathcal{P}^t)P(j^\star) \leq \frac{P\tau_0}{CS}.$$

*This completes the proofs.*

Specifically, because $\mathcal{C}$ only differs from $\hat{\mathcal{C}}$ on client $i^\star$, we could bound each term on two different situations respectively.

We consider the difference of the local updates on the client $i$ ($i \neq i^\star$).

**Lemma 5** *Under Assumption 1, we can bound the difference of the local updates on the active client $i$ ($i \neq i^\star$). The local update satisfies:*

$$\mathbb{E}\| \left( \theta_{i,k+1}^t - \theta^t \right) - \left( \hat{\theta}_{i,k+1}^t - \hat{\theta}^t \right) \| \leq \eta^t K L \mathbb{E}\|\theta^t - \hat{\theta}^t\| + \eta^t K \mathbb{E}\|\lambda_i^t - \hat{\lambda}_i^t\|. \qquad (17)$$

**Proof.** *Reconstructing Eq.(13) we can get the following iteration relationship:*

$$\theta_{i,k+1}^t - \theta^t = \left( 1 - \eta^t \rho \right) \left( \theta_{i,k}^t - \theta^t \right) - \eta^t \left( g_{i,k}^t + \lambda_i^t \right).$$

*On each client $i$ ($i \neq i^\star$), each data sample is the same, thus we have:*

$$\mathbb{E}\| \left( \theta_{i,k+1}^t - \theta^t \right) - \left( \hat{\theta}_{i,k+1}^t - \hat{\theta}^t \right) \|$$

$$= \mathbb{E}\| \left( 1 - \eta^t \rho \right) \left[ \left( \theta_{i,k}^t - \theta^t \right) - \left( \hat{\theta}_{i,k}^t - \hat{\theta}^t \right) \right] - \eta^t \left( g_{i,k}^t - \hat{g}_{i,k}^t \right) - \eta^t \left( \lambda_i^t - \hat{\lambda}_i^t \right) \|$$

$$\leq \left( 1 - \eta^t \rho \right) \mathbb{E}\| \left( \theta_{i,k}^t - \theta^t \right) - \left( \hat{\theta}_{i,k}^t - \hat{\theta}^t \right) \| + \eta^t L \mathbb{E}\|\theta_{i,k}^t - \hat{\theta}_{i,k}^t\| + \eta^t \mathbb{E}\|\lambda_i^t - \hat{\lambda}_i^t\|$$

$$\leq \left( 1 - \eta^t \rho_L \right) \mathbb{E}\| \left( \theta_{i,k}^t - \theta^t \right) - \left( \hat{\theta}_{i,k}^t - \hat{\theta}^t \right) \| + \eta^t L \mathbb{E}\|\theta^t - \hat{\theta}^t\| + \eta^t \mathbb{E}\|\lambda_i^t - \hat{\lambda}_i^t\|.$$

*where $\rho_L = \rho - L$ is a constant.*

*Unrolling the recursion from $k = 0$ to $K - 1$, and adopting the factors $\theta_i^{t+1} = \theta_{i,k}^t$ and $\theta_{i,0}^t = \theta^t$, we have:*

$$\mathbb{E}\| \left( \theta_i^{t+1} - \theta^t \right) - \left( \hat{\theta}_i^{t+1} - \hat{\theta}^t \right) \| = \mathbb{E}\| \left( \theta_{i,k}^t - \theta^t \right) - \left( \hat{\theta}_{i,k}^t - \hat{\theta}^t \right) \|$$

$$\leq \left[ \prod_{k=0}^{K-1} \left( 1 - \eta^t \rho_L \right) \right] \mathbb{E}\| \left( \theta_{i,0}^t - \theta^t \right) - \left( \hat{\theta}_{i,0}^t - \hat{\theta}^t \right) \|$$

$$+ \sum_{k=0}^{K-1} \eta^t \left[ \prod_{j=k+1}^{K-1} \left( 1 - \eta^t \rho_L \right) \right] \left( L \mathbb{E}\|\theta^t - \hat{\theta}^t\| + \mathbb{E}\|\lambda_i^t - \hat{\lambda}_i^t\| \right)$$

$$= \sum_{k=0}^{K-1} \eta^t \left[ \prod_{j=k+1}^{K-1} \left( 1 - \eta^t \rho_L \right) \right] \left( L \mathbb{E}\|\theta^t - \hat{\theta}^t\| + \mathbb{E}\|\lambda_i^t - \hat{\lambda}_i^t\| \right).$$

*Simplifying the relationships, we have:*

$$\mathbb{E}\| \left( \theta_i^{t+1} - \theta^t \right) - \left( \hat{\theta}_i^{t+1} - \hat{\theta}^t \right) \| = \frac{1 - \left( 1 - \eta^t \rho_L \right)^K}{\rho_L} \left( L \mathbb{E}\|\theta^T - \hat{\theta}^T\| + \mathbb{E}\|\lambda_i^t - \hat{\lambda}_i^t\| \right)$$

$$\leq \eta^t K L \mathbb{E}\|\theta^T - \hat{\theta}^T\| + \eta^t K \mathbb{E}\|\lambda_i^t - \hat{\lambda}_i^t\|.$$

*The last inequality adopts the Bernoulli inequality $(1 + x)^K \geq 1 + Kx$ for $K \geq 1$ and $x \geq -1$.*

Then we consider the difference of the local updates on client $i^\star$.

**Lemma 6** *Under Assumption 1 and 2, we can bound the difference of the local updates on the active client $i^\star$. The local update satisfies:*

$$\mathbb{E}\| \left( \theta_{i,k+1}^t - \theta^t \right) - \left( \hat{\theta}_{i,k+1}^t - \hat{\theta}^t \right) \| \leq \eta^t K L \mathbb{E}\|\theta^t - \hat{\theta}^t\| + \eta^t K \mathbb{E}\|\lambda_i^t - \hat{\lambda}_i^t\| + \frac{2\eta^t K G}{s}. \qquad (18)$$

**Proof.** *Reconstructing Eq.(13) we can get the following iteration relationship:*

$$\theta_{i,k+1}^t - \theta^t = \left( 1 - \eta^t \rho \right) \left( \theta_{i,k}^t - \theta^t \right) - \eta^t \left( g_{i,k}^t + \lambda_i^t \right).$$

*Lemma 5 shows the recursive formulation when we select the same data sample. However, on the client $i^\star$, it also may select the different sample pairs. Therefore, we first study the recursive*

*formulation of this situation. When the stochastic gradients are calculated with different sample pairs* $(\xi, \hat{\xi})$, *we have:*

$$\mathbb{E}\| \left( \theta_{i,k+1}^t - \theta^t \right) - \left( \hat{\theta}_{i,k+1}^t - \hat{\theta}^t \right) \|$$

$$= \mathbb{E}\| \left( 1 - \eta^t \rho \right) \left[ \left( \theta_{i,k}^t - \theta^t \right) - \left( \hat{\theta}_{i,k}^t - \hat{\theta}^t \right) \right] - \eta^t \left( g_{i,k}^t - \hat{g}_{i,k}^t \right) - \eta^t \left( \lambda_i^t - \hat{\lambda}_i^t \right) \|$$

$$\leq \left( 1 - \eta^t \rho \right) \mathbb{E}\| \left( \theta_{i,k}^t - \theta^t \right) - \left( \hat{\theta}_{i,k}^t - \hat{\theta}^t \right) \| + 2\eta^t G + \eta^t \mathbb{E}\|\lambda_i^t - \hat{\lambda}_i^t\|.$$

*For every single sample, the probability of selecting the* $\xi_{j^\star}$ *is* $\frac{1}{S}$. *Therefore, combining Lemma 5, we have:*

$$\mathbb{E}\| \left( \theta_{i,k+1}^t - \theta^t \right) - \left( \hat{\theta}_{i,k+1}^t - \hat{\theta}^t \right) \|$$

$$\leq \left( 1 - \frac{1}{S} \right) \left[ \left( 1 - \eta^t \rho_L \right) \mathbb{E}\| \left( \theta_{i,k}^t - \theta^t \right) - \left( \hat{\theta}_{i,k}^t - \hat{\theta}^t \right) \| + \eta^t L \mathbb{E}\|\theta^t - \hat{\theta}^t\| + \eta^t \mathbb{E}\|\lambda_i^t - \hat{\lambda}_i^t\| \right]$$

$$+ \frac{1}{S} \left[ \left( 1 - \eta^t \rho \right) \mathbb{E}\| \left( \theta_{i,k}^t - \theta^t \right) - \left( \hat{\theta}_{i,k}^t - \hat{\theta}^t \right) \| + 2\eta^t G + \eta^t \mathbb{E}\|\lambda_i^t - \hat{\lambda}_i^t\| \right]$$

$$\leq \left( 1 - \eta^t \rho_L \right) \mathbb{E}\| \left( \theta_{i,k}^t - \theta^t \right) - \left( \hat{\theta}_{i,k}^t - \hat{\theta}^t \right) \| + \eta^t L \mathbb{E}\|\theta^t - \hat{\theta}^t\| + \eta^t \mathbb{E}\|\lambda_i^t - \hat{\lambda}_i^t\| + \frac{2\eta^t G}{s}.$$

*Generally, we consider the size of samples* $S$ *to be large enough. In current deep learning, the dataset adopted usually maintains even millions of samples, which indicates that* $1 - \frac{1}{S} \to 1$.

*Unrolling the recursion from* $k = 0$ *to* $K - 1$ *and adopting the* $\theta_i^{t+1} = \theta_{i,K}^t$ *and* $\theta_{i,0}^t = \theta^t$, *we have a similar relationship:*

$$\mathbb{E}\| \left( \theta_i^{t+1} - \theta^t \right) - \left( \hat{\theta}_i^{t+1} - \hat{\theta}^t \right) \| = \mathbb{E}\| \left( \theta_{i,k}^t - \theta^t \right) - \left( \hat{\theta}_{i,k}^t - \hat{\theta}^t \right) \|$$

$$\leq \sum_{k=0}^{K-1} \eta^t \left[ \prod_{j=k+1}^{K-1} \left( 1 - \eta^t \rho_L \right) \right] \left( L \mathbb{E}\|\theta^t - \hat{\theta}^t\| + \mathbb{E}\|\lambda_i^t - \hat{\lambda}_i^t\| + \frac{2G}{s} \right).$$

*Simplifying the relationships, we have:*

$$\mathbb{E}\| \left( \theta_i^{t+1} - \theta^t \right) - \left( \hat{\theta}_i^{t+1} - \hat{\theta}^t \right) \| = \frac{1 - \left( 1 - \eta^t \rho_L \right)^K}{\rho_L} \left( L \mathbb{E}\|\theta^t - \hat{\theta}^t\| + \mathbb{E}\|\lambda_i^t - \hat{\lambda}_i^t\| + \frac{2G}{s} \right)$$

$$\leq \eta^t K L \mathbb{E}\|\theta^t - \hat{\theta}^t\| + \eta^t K \mathbb{E}\|\lambda_i^t - \hat{\lambda}_i^t\| + \frac{2\eta^t K G}{s}.$$

*The last inequality adopts the Bernoulli inequality.*

### B.3.2 Proofs

Table 8: Notations in the proofs.

| Symbol | Formulation | Description |
|--------|-------------|-------------|
| $\Delta^t$ | $\mathbb{E}\|\theta^t - \hat{\theta}^t\|$ | difference of the global parameters |
| $\delta^t$ | $\frac{1}{C} \sum_{i \in \mathcal{C}} \mathbb{E}\|\theta_i^t - \hat{\theta}_i^t\|$ | discrete difference of the local parameters |
| $\sigma^t$ | $\frac{1}{C} \sum_{i \in \mathcal{C}} \mathbb{E}\|\lambda_i^t - \hat{\lambda}_i^t\|$ | discrete difference of the dual variables |
| $\pi^t$ | $\frac{1}{C} \sum_{i \in \mathcal{C}} \mathbb{E}\|(\theta_i^t - \theta^{t-1}) - (\hat{\theta}_i^t - \hat{\theta}^{t-1})\|$ | discrete difference of the local updates |

In this part, we mainly introduce the proof of the main theorems. Combining the local updates and global updates, we can further upper bound both the primal and dual variables. Before proving the theorems, we first introduce the notations of updates of the global parameters and dual variables in Table 8.

$\Delta^t$ measures the difference of the primal models during the training. $\delta^t$ is the local separate difference when the local objective is solved. $\sigma^t$ measures the difference of the dual gaps. $\pi^t$ measures the

difference of the local updates, which is also an important variable to connect global variables and dual variables. In our proofs, we first discuss the process of the primal variables and dual variables respectively. Then we can use the $\pi$ term to construct an inequality to eliminate redundant terms, which could further provide the recursive relationship of the primal variables and dual variables separately. Next, we introduce these three processes one by one.

From the global updates, according to the global aggregation and let $\mathbb{I}_i$ be the indicator function, we have:

$$
\begin{aligned}
\Delta^{t+1} = \mathbb{E}\|\theta^{t+1} - \hat{\theta}^{t+1}\| &= \mathbb{E}\|\frac{1}{P}\sum_{i \in \mathcal{P}^t}\left(\theta_i^{t+1} - \hat{\theta}_i^{t+1}\right) + \frac{1}{\rho}\left(\overline{\lambda}_i^{t+1} - \hat{\overline{\lambda}}_i^{t+1}\right)\| \\
&\le \frac{1}{P}\mathbb{E}\sum_{i \in \mathcal{C}}\mathbb{E}\|\theta_i^{t+1} - \hat{\theta}_i^{t+1}\| \cdot \mathbb{I}_i + \frac{1}{\rho}\mathbb{E}\|\overline{\lambda}_i^{t+1} - \hat{\overline{\lambda}}_i^{t+1}\| \\
&\le \frac{1}{C}\sum_{i \in \mathcal{C}}\mathbb{E}\|\theta_i^{t+1} - \hat{\theta}_i^{t+1}\| + \frac{1}{C}\sum_{i \in \mathcal{C}}\frac{1}{\rho}\mathbb{E}\|\overline{\lambda}_i^{t+1} - \hat{\overline{\lambda}}_i^{t+1}\| \le \delta^{t+1} + \frac{1}{\rho}\sigma^{t+1}.
\end{aligned}
$$

From the dual updates, according to the local dual variable, we have two different cases. Combing the *D-Update* we have:

$$
\begin{aligned}
\overline{\lambda}^{t+1} &= \frac{1}{C}\sum_{i \in \mathcal{C}}\lambda_i^{t+1} = \frac{1}{C}\sum_{i \in \mathcal{C}}\lambda_i^t + \frac{1}{C}\sum_{i \in \mathcal{P}^t}\rho(\theta_i^{t+1} - \theta^t) + \frac{1}{C}\sum_{i \notin \mathcal{P}^t}\rho(\overline{\theta}^{t+1} - \theta^t) \\
&= \frac{1}{C}\sum_{i \in \mathcal{C}}\lambda_i^t + \frac{P}{C}\rho(\overline{\theta}^{t+1} - \theta^t) + \frac{C-P}{C}\rho(\overline{\theta}^{t+1} - \theta^t) = \overline{\lambda}^t + \rho(\overline{\theta}^{t+1} - \theta^t).
\end{aligned}
$$

Considering the randomness of selecting $\mathcal{P}^t$ and expectation of $\overline{\theta}$, we have $\sigma^{t+1} \le \sigma^t + \rho\pi^t$.

Here we add the additional definition of the unparticipated clients. We let the $\theta_i^{t+1} = \theta^t$ where $i \notin \mathcal{P}^t$, which enlarge the summation from $\mathcal{P}^t$ to $\mathcal{C}$. Then we summarize Lemma 5 and 6 as the following formulation:

$$
\begin{aligned}
\pi^{t+1} &= \frac{1}{C}\sum_{i \in \mathcal{C}}\mathbb{E}\|\left(\theta_i^{t+1} - \theta^t\right) - \left(\hat{\theta}_i^{t+1} - \hat{\theta}^t\right)\| = \frac{1}{C}\sum_{i \in \mathcal{P}^t}\mathbb{E}\|\left(\theta_i^{t+1} - \theta^t\right) - \left(\hat{\theta}_i^{t+1} - \hat{\theta}^t\right)\| \\
&< \eta^t KL\mathbb{E}\|\theta^t - \hat{\theta}^t\| + \eta^t K\frac{1}{C}\sum_{i \in \mathcal{C}}\mathbb{E}\|\lambda_i^t - \hat{\lambda}_i^t\| + \frac{2\eta^t KG}{CS} = c_1\Delta^t + c_2\sigma^t + c_3.
\end{aligned}
$$

Finally, we can directly expand the $\pi$ term by the triangle inequality:

$$
\begin{aligned}
\delta^{t+1} &= \frac{1}{C}\sum_{i \in \mathcal{C}}\mathbb{E}\|\theta_i^{t+1} - \hat{\theta}_i^{t+1}\| \\
&\le \frac{1}{C}\sum_{i \in \mathcal{C}}\mathbb{E}\|\left(\theta_i^{t+1} - \theta^t\right) - \left(\hat{\theta}_i^{t+1} - \hat{\theta}^t\right)\| + \frac{1}{C}\sum_{i \in \mathcal{C}}\mathbb{E}\|\theta^t - \hat{\theta}^t\| = \pi^{t+1} + \Delta^t.
\end{aligned}
$$

Combing the above recursive formulations, we have:

$$
\begin{cases}
\Delta^{t+1} & \le \delta^{t+1} + \frac{1}{\rho}\sigma^{t+1}, \\
\sigma^{t+1} & \le \sigma^t + \rho\pi^{t+1}, \\
\delta^{t+1} & \le \pi^{t+1} + \Delta^t, \\
\pi^{t+1} & \le c_1\Delta^t + c_2\sigma^t + c_3.
\end{cases}
$$

By multiplying three additional positive coefficients $\alpha$, $\beta$, and $\gamma$ on the last three inequalities respectively, and adding them to the first one, we have:

$$
\Delta^{t+1} + \left(\alpha - \frac{1}{\rho}\right)\sigma^{t+1} + (\beta - 1)\delta^{t+1} + (\gamma - \alpha\rho - \beta)\pi^{t+1} \le (\beta + \gamma c_1)\Delta^t + (\alpha + \gamma c_2)\sigma^t + \gamma c_3.
$$

By observing the LHS and RHS of the inequality, we notice that it could be summarized as a recursive formulation of $\Delta^t$ and $\sigma^t$ terms by selecting proper coefficients. Therefore, let the following conditions hold,

$$\beta - 1 \geq 0,$$
$$\gamma - \alpha\rho - \beta \geq 0.$$

By simply selecting the minimal values of $\beta = 1$ and $\gamma = 1 + \alpha\rho$, we have:

$$\Delta^{t+1} + \left(\alpha - \frac{1}{\rho}\right)\sigma^{t+1} = \Delta^{t+1} + \left(\alpha - \frac{1}{\rho}\right)\sigma^{t+1} + (\beta - 1)\delta^{t+1} + (\gamma - \alpha\rho - \beta)\pi^{t+1}$$

$$\leq (\beta + \gamma c_1)\Delta^t + (\alpha + \gamma c_2)\sigma^t + \gamma c_3$$

$$\leq (1 + \gamma\eta^t KL)\left(\Delta^t + \frac{\alpha + (1 + \alpha\rho)\eta^t K}{1 + (1 + \alpha\rho)\eta^t KL}\right) + \frac{2\gamma\eta^t KG}{CS}.$$

Here we further let $\alpha - \frac{1}{\rho} \geq \frac{\alpha + (1+\alpha\rho)\eta^t K}{1 + (1+\alpha\rho)\eta^t KL}$ to support the above recursive relationships. To satisfy this, we can solve the following inequality:

$$L(\alpha\rho)^2 - \rho(\alpha\rho) - \left(L + \rho + \frac{1}{\eta^t K}\right) \geq 0.$$

From the fundamental knowledge of quadratic equations, we know that there must exist a positive number belonging to interval $[x^+, +\infty]$ to satisfy the condition, where $x^+$ is the larger zero point of the equation $Lx^2 - \rho x - \left(L + \rho + \frac{1}{\eta^t K}\right) = 0$. We can solve $x^+ = \frac{\rho + \sqrt{\rho^2 + 4L(L+\rho+\frac{1}{\eta^t K})}}{2L} = \frac{\rho + \sqrt{(\rho+2L)^2 + \frac{4L}{\eta^t K}}}{2L} \leq 1 + \frac{\rho}{L} + 2\sqrt{\frac{L}{\eta^t K}}$. When we select the proper $\alpha\rho \geq x^+$, the above inequality always holds. This also indicates that $\alpha - \frac{1}{\rho} \geq \frac{x^+ - 1}{\rho} > \frac{1}{L}$. Here we denote $\alpha_\rho = \alpha - \frac{1}{\rho}$ and the previous definition $\gamma = 1 + \alpha\rho$ as two constant coefficients, we can simplify the final iteration relationship as:

$$\Delta^{t+1} + \alpha_\rho\sigma^{t+1} \leq \left(1 + \gamma\eta^t KL\right)\left(\Delta^t + \alpha_\rho\sigma^t\right) + \frac{\gamma\eta^t KG}{CS}.$$

Unrolling this from $t = \tau_0$ to $T - 1$ and adopting the factors of $\Delta^{\tau_0} = 0$ and $\lambda^{\tau_0} = 0$, we have:

(1) When the global learning rate is selected as a constant $\eta^t = \eta_0^t$ where the initial learning rate $\eta_0^t \leq \frac{1}{KL}$:

$$\Delta^T + \alpha_\rho\lambda^T \leq \sum_{t=\tau_0}^{T-1}\left(1 + \gamma\eta_0^t KL\right)^t \frac{2\gamma\eta_0^t KG}{CS} < \frac{2G(1 + \gamma)^T}{LCS}.$$

(2) When the global learning rate is selected as a decayed sequence $\eta^t = \frac{\eta_0}{t+1}$ where the initial learning rate $\eta_0 \leq \frac{\mu}{\gamma K}$ where $\mu$ is a positive constant, we have:

$$\Delta^T + \alpha_\rho\lambda^T \leq \sum_{t=\tau_0}^{T-1}\left(\prod_{j=t}^{T-1}\left(1 + \frac{\gamma\eta_0^t KL}{j+1}\right)\right)\frac{2\gamma\eta_0^t KG}{CS(t+1)} \leq \sum_{t=\tau_0}^{T-1}\exp\left(\gamma\eta_0^t KL\ln\left(\frac{T}{t+1}\right)\right)\frac{2\gamma\eta_0^t KG}{CS(t+1)}$$

$$= \frac{2\gamma\eta_0^t KGT^{\gamma\eta_0^t KL}}{CS}\sum_{t=\tau_0}^{T-1}\left(\frac{1}{t+1}\right)^{1+\gamma\eta_0^t KL} < \frac{2G}{LCS}\left(\frac{T}{\tau_0}\right)^{\mu L}.$$

Here we mainly focus on the case of the decayed learning rates. According to Lemma 4, we have:

$$\varepsilon_G \leq G\Delta^T + \frac{HP\tau_0}{CS} \leq \frac{2G^2}{LCS}\left(\frac{T}{\tau_0}\right)^{\mu L} + \frac{HP\tau_0}{CS}.$$

By selecting the proper $\tau_0 = \left(\frac{2G^2}{HPL}\right)^{\frac{1}{1+\mu L}}T^{\frac{\mu L}{1+\mu L}}$, we can get the minimal error bound as:

$$\varepsilon_G \leq \frac{2}{CS}\left(\frac{2G^2}{L}\right)^{\frac{1}{1+\mu L}}(HPT)^{\frac{\mu L}{1+\mu L}}.$$

**Discussions of the generalization errors.** We show some general results of the generalization errors in the following Table 9. We prove that the federated primal-dual family can benefit from the local interval $K$ than the vanilla SGD methods, which is one of the key properties of the primal-dual methods.

Table 9: Generalization error bounds of smooth non-convex objectives.

|  | Assumption | Generalization |
|---|---|---|
| Hardt et al. [2016] | Lipschitz | $\mathcal{O}(\frac{(KT)^{\frac{\mu L}{1+\mu L}}}{CS})$ |
| Mohri et al. [2019] | Lipschitz, VC | $\mathcal{O}(\frac{TK}{\sqrt{CS}})$ |
| Hu et al. [2022] | Lipschitz, Bernstein | $\mathcal{O}(\frac{TK}{\sqrt{CS}})$ |
| Wu et al. [2023] | Lipschitz, Stochastic | $\mathcal{O}(\frac{\sqrt{T}}{C\sqrt{CS}})$ |
| Sun et al. [2023d] | Lipschitz | $\mathcal{O}(\frac{(PKT)^{\frac{\mu L}{1+\mu L}}}{CS})$ |
| our | Lipschitz | $\mathcal{O}(\frac{(PT)^{\frac{\mu L}{1+\mu L}}}{CS})$ |

