# OpenReview forum: "A-FedPD: Aligning Dual-Drift is All Federated Primal-Dual Learning Needs"
_NeurIPS.cc/2024/Conference — NeurIPS 2024 spotlight_

### Official Review · Reviewer_c2hH · 2024-06-28

**Soundness:** 3
**Presentation:** 2
**Contribution:** 2
**Rating:** 6
**Confidence:** 4

**Summary:**

The authors propose an algorithm for distributed optimization of a sum of nonconvex smooth functions, with partial participation. They obtain a O(1/T) rate.

**Strengths:**

The study of convergence of algorithms adapted to federated learning is an important topic.

**Weaknesses:**

* There is no discussion of the O(1/T) rate in Theorem 1 with respect to existing results, for instance on Scaffold. I don't think the method improves on the SOTA.
* In Federated Learning (FL), the bottleneck is communication. The complexity in number of communication rounds is only one part of it, sending full vectors is not reasonable in FL, compression must be used to reduce dramatically the number of communicated bits, which is the right criterion to study.
* There is a lack of comparison to existing methods.  Here the local steps are performed to find a stationary point of the augmented Lagrangian. This is equivalent to compute, inexactly, a proximity operator. This is exactly what the 5GCS algorithm does, proposed in
Grudzien et al. "Can 5th generation local training methods support client sampling? yes!" AISTATS 2023. 5GCS has been extended to compression in Grudzien et al "Improving Accelerated Federated Learning with Compression and Importance Sampling", arXiv:2306.03240. Another technique is loopless: instead of an inner loop of local steps, communication is triggered randomly. This is the technique used in Scaffnew, an important algorithm since which demonstrates that local training yields acceleration of communication in FL: Mishchenko et al. "ProxSkip: Yes! Local gradient steps provably lead to communication acceleration! Finally!" ICML 2022. Scaffnew has been interpreted as a primal-dual algorithm in Condat and Richtárik, "RandProx: Primal-Dual Optimization Algorithms with Randomized Proximal Updates" ICLR 2023. Scaffnew has been extended to partial participation, and also compression, as TAMUNA in Condat et al. "TAMUNA: Doubly Accelerated Federated Learning with Local Training, Compression, and Partial Participation" arXiv:2302.09832.

* In these algorithms, such as 5GCS and TAMUNA, the dual variables of the inactive clients are not modified. This is fine, because the aggregated model from the active clients does not contain any information about the inactive clients, so we cannot expect to use the new model to update these dual variables. So, I don't see what the challenge addressed in the paper is, because combining local training and partial participation can been done successfully, as these papers show.

* The writing is not good. The first 3 pages contain only vague statements, without the problem being defined. The paper contains a lot of unusual or weird expressions, such as "imminent challenge" "brings great distress for training". What does the sentence "Ozfatura et al. [2021], Xu et al. [2021] propose to adopt the global update consistently control the local training to force a high consensus." mean?
* I don't see what "dual drift" actually means.  If the dual variables of inactive clients remain the same for a long time, this is not a drift. The only explanation of this notion is that there is "excessive differences between the primal and dual variables", which is unclear, one cannot compare a primal and dual variable. If you mean that the primal and dual variables become inconsistent with each other, this inconsistency should be made precise. So, the paper title should be modified. Also, it is not grammatically correct, this should be "Needs", not "Need".




Typos

* (5) first update: this should be $\theta_i$, not $\theta_i^t$

* Algorithm 1, line 4: this should be $P^t$, not $N^t$

**Questions:**

* Theorem 1 "let rho > O(L^3) be positive with lower bound": what does it mean ?

* rho proportional to $L^3$ is weird, the left hand side and right hand side of (8) are not homogeneous. Looking at the proof, Line 987: is is not clear at all what the condition on rho is.

* line 3: "randomly select active clients set" : do you mean choose a subset of size P uniformly at random?

* Why does the number P of active clients not appear in Theorem 1? This is strange.

* Line 972: how do you get the factors 4 and the 2, this looks like Young's inequality but not clear at all.

---

> ### Author Rebuttal · Authors · 2024-08-04
>
> **W1 and W2:  I don't think the method improves on the SOTA and compression must be used to reduce dramatically the number of communicated bits, which is the right criterion to study.**
>
> The main contribution of this paper is neither proposing a theory that achieves faster optimization rates than SOTA nor introducing a compression technique to reduce communication bits. Moreover, we have never claimed in this paper that our method can achieve a faster rate than the SOTA  results. **The primary focus of this paper is on addressing the training instability issues encountered in experiments of existing primal-dual methods.** Our research is not directly related to these two fields. We introduce our main contribution **in the overall response above**.
>
> **W3: There is a lack of comparison to existing methods.**
>
> Thank you very much for pointing out these excellent works. We will cite these theory-related works in section 2. We also summarize the differences between these works and ours **in the overall response above**.
>
> **W4 and W6: I don't see what "dual drift" actually means.**
>
> Thank you for this concern. Since this is the main issue studied in this paper, we specifically introduce this **in the overall response above**.
>
> **W5: The writing is not good.**
>
> We are sorry if some sentences may have caused misunderstandings. We will clarify their expressions and revise them to make them more straightforward and understandable.
>
> **Q1 and Q2: Theorem 1 "let rho > O(L^3) be positive with lower bound": what does it mean? Looking at the proof, Line 987: it is not clear at all what the condition on rho is.**
>
> Thank you for this question and we will revise this ambiguous expression.
>
> **Is $\rho$ proportional to $L^3$ ?**
> No. $\rho$ is just a general constant. We follow the proof techniques in [X1] to construct the recurrsive relationship on the combination of $\mathbb{E}[f(\overline{\theta}^t)]$ term and $R_t=\frac{1}{C}\mathbb{E}\Vert\theta_i^{t+1}-\overline{\theta}^t\Vert$ term. We can construct an arithmetic sequence when the following condition is satisfied (line 987 in our paper):
> $$
> \rho^2 - 4L^3\rho - 36L^2-4L^4>0.
> $$
> This can be regarded as a quadratic function, using the basic knowledge of the roots of quadratic functions, we have:
> $$
> \rho > \frac{4L^3 + \sqrt{16L^6 + 16L^4 + 144L^2}}{2} \ \ \text{or} \ \ \rho<\frac{4L^3 - \sqrt{16L^6 + 16L^4 + 144L^2}}{2}.
> $$
> Since $4L^3 - \sqrt{16L^6 + 16L^4 + 144L^2}<0$ but we need $\rho$ to be positive. Therefore, the required range of $\rho$ is $\rho > \frac{4L^3 + \sqrt{16L^6 + 16L^4 + 144L^2}}{2}$ only.
>
> Sorry for using an ambiguous expression here and $\rho=\mathcal{O}(L^3)$ is different from our $\rho>\mathcal{O}(L^3)$. The form of this constant is quite cumbersome when written in the theorem. Therefore, we used $O(L^3)$ as a placeholder, which has caused some ambiguity. We apologize for this and will correct it to $\rho > \frac{4L^3 + \sqrt{16L^6 + 16L^4 + 144L^2}}{2}$.
>
> [X1] Acar D A E, Zhao Y, Matas R, et al. Federated Learning Based on Dynamic Regularization[C]//International Conference on Learning Representations.
>
> **Q3: line 3: "randomly select active clients set": do you mean to choose a subset of size P uniformly at random?**
>
> It means that before each round of local training begins, the server will activate a batch of local clients to participate in the training, with each client having the same probability of being activated. After completing the training, they will remain inactive until they are selected to be activated again.
>
> **Q4: Why does the number P of active clients not appear in Theorem 1? This is strange.**
>
> Virtual updates can be understood as a form of synchronization for non-participating clients, and can also be seen as a form of virtual full-participation training. The conclusions drawn from this approach would not exceed those given in FedPD [X2], as the algorithmic proof in FedPD requires strict full participation. We present the comparison of the theoretical results in Table 7. For FedPD, even in the case of full participation, it does not indicate that the results are related to $C$. For FedDyn, the proofs indicate that the final convergence rate achieves $\mathcal{O}(\frac{C}{PT})$ (slower than $\mathcal{O}(\frac{1}{T})$). Our analysis shows that, with virtual updates and appropriate parameter choices to avoid excessive errors, the convergence rate of partial participation can achieve a similar complexity as full participation, thus reducing the impact of the $\frac{C}{P}$ constant term in FedDyn.
>
> [X2] Zhang X, Hong M, Dhople S, et al. Fedpd: A federated learning framework with adaptivity to non-iid data[J]. IEEE Transactions on Signal Processing, 2021, 69: 6055-6070.
>
> **Q5: Line 972: how do you get the factors 4 and the 2, this looks like Young's inequality but not clear at all.**
>
> Thank you for pointing out this.
>
> This is the noised version of proof of Lemma 11 in [X1] (page 35 in their paper). The coefficients $4$ and $2$ in our conclusion are consistent with those in their proof. This is because the proof in [X1] uses Young's inequality to first divide these terms into four groups, thus enlarging the coefficients by a factor of $4$. They then merge the dual updates and local update terms into a single upper bound, resulting in the constant $2$. Our conclusion additionally considers a case with local error, which introduces a constant $\epsilon$ due to the local inexact solution. We apologize for the lack of clarity here. **We will directly copy the lemma they used and cite it as their Lemma 11 in the next version. We hope this resolves your concerns.**
>
> [X1] Acar D A E, Zhao Y, Matas R, et al. Federated Learning Based on Dynamic Regularization[C]//International Conference on Learning Representations.
>
> Thanks for reading our rebuttal and we are happy to continue discussions if there are some other concerns unsolved.

---

> > ### Comment · Reviewer_c2hH · 2024-08-08
> >
> > Papers with strong theoretical contributions and papers with heuristics showing strong empirical performance are both valuable. Also, nonconvex optimization is more difficult than convex optimization. I understand that you propose a heuristic technique to mitigate the negative effects of partial participation in nonconvex federated learning. You motivate your technique by an intuition based on correcting the "dual drift". My negative evaluation is based on the fact that there is a theoretically-grounded literature for the convex case, and it seems to contradict your intuition that it is bad if the dual variables of idle clients remain unchanged for a long time. In other words, your technique should be shown to work in the "simple" convex case, which is not the case. How do we know that A-FedPD will not diverge in even simple quadratic synthetic experiments? Showing improved generalization efficiency in some practical examples is fine, but this is not enough to make progress in our understanding of primal-dual optimization methods for federated learning, as the title claims, in my opinion.
> > At this time, I am keeping my score.

---

> > > ### Author Response · Authors · 2024-08-08
> > > **Rebuttal from authors**
> > >
> > > Thank you for your positive response and we are honored and happy to continue the discussion with you. Regarding your comments above, we have listed and addressed the two concerns you raised below.
> > >
> > > ## About the comment "there is a theoretically-grounded literature for the convex case seems to contradict your intuition".
> > >
> > > In our overall response, we have made it very clear that even for non-convex objectives, the classical federated primal-dual method FedADMM with partial participation have been theoretically proven to converge [X1]. However, despite the theoretical guarantees, our experiments have validated that FedADMM still faces extreme instability and even divergence during partial participation training (our figure 1). Obviously, **pure theoretical analysis can not provide an absolute guarantee** of an algorithm's feasibility. Therefore, it is not appropriate to dismiss the existence of the problem simply because it has been proven in theory. In fact, we are not the first to observe such non-convergence phenomena in experiments; previous classical works [X2,X3,X4] have repeatedly demonstrated the existence of this phenomenon experimentally. [X4] also indicates that the update mismatch between primal and dual variables leads to a "drift", which is related to the ``dual drift" we pointed out in this paper.
> > >
> > > [X1] Wang H, Marella S, Anderson J. Fedadmm: A federated primal-dual algorithm allowing partial participation[C]//2022 IEEE 61st Conference on Decision and Control (CDC). IEEE, 2022: 287-294.
> > >
> > > [X2] Xu J, Wang S, Wang L, et al. Fedcm: Federated learning with client-level momentum[J]. arXiv preprint arXiv:2106.10874, 2021.
> > >
> > > [X3] Baumgart G A, Shin J, Payani A, et al. Not All Federated Learning Algorithms Are Created Equal: A Performance Evaluation Study[J]. arXiv preprint arXiv:2403.17287, 2024.
> > >
> > > [X4] Kang H, Kim M, Lee B, et al. FedAND: Federated Learning Exploiting Consensus ADMM by Nulling Drift[J]. IEEE Transactions on Industrial Informatics, 2024.
> > >
> > > ## About the comment "Your technique should be shown to work in the "simple" convex case".
> > >
> > > We cannot agree with this comment. As you mentioned above, nonconvex optimization is more difficult than convex optimization. In fact, **a large number of papers on learning federated primal-dual methods conduct the experiments on non-convex experiments [X1, X2, X3, X4, X5]**.
> > >
> > > More importantly, **the "dual drift" issue is discovered in non-convex experiments** by several previous works (mentioned in the answer above), and our technique is aimed at addressing this problem. We have not identified this issue in convex experiments. Therefore, the reviewers have no reason to ask us to implement this work for convex objectives. If the ``dual drift" issue does not indeed exist in convex experiments, then our virtual update technique is not necessary for training convex models. However, since a lot of studies have identified this problem in non-convex experiments, we have validated that our method effectively addresses this issue in non-convex experiments.
> > >
> > > [X1] Zhang X, Hong M, Dhople S, et al. Fedpd: A federated learning framework with adaptivity to non-iid data[J]. IEEE Transactions on Signal Processing, 2021, 69: 6055-6070.
> > >
> > > [X2] Acar D A E, Zhao Y, Matas R, et al. Federated Learning Based on Dynamic Regularization[C]//International Conference on Learning Representations.
> > >
> > > [X3] Sun Y, Shen L, Huang T, et al. FedSpeed: Larger Local Interval, Less Communication Round, and Higher Generalization Accuracy[C]//The Eleventh International Conference on Learning Representations.
> > >
> > > [X4] Kang H, Kim M, Lee B, et al. FedAND: Federated Learning Exploiting Consensus ADMM by Nulling Drift[J]. IEEE Transactions on Industrial Informatics, 2024.
> > >
> > > [X5] Zhang Y, Tang D. A differential privacy federated learning framework for accelerating convergence[C]//2022 18th International Conference on Computational Intelligence and Security (CIS). IEEE, 2022: 122-126.
> > >
> > > We noted that the reviewers show concerns about the naming. One is with the term "dual drift", and the other is with the title. We believe these are very easy to resolve. We would greatly appreciate any suggestions the reviewers might have for the names.

---

> > > > ### Comment · Reviewer_c2hH · 2024-08-09
> > > >
> > > > I am happy to participate to this polite and argued discussion.
> > > >
> > > > 1) I don't understand your reply. If FedADMM is proved to converge (in nonconvex problems with partial participation), it is not possible that it diverges in practice (unless too agressive stepsizes are chosen, but this is normal).
> > > >
> > > > 2) I see, this is interesting if the dual drift issue exists in the nonconvex setting but not in the convex setting. I did not request you to study the convex setting, I was just not convinced by the dual drift issue that you are considering in the first place, precisely because I am familiar with the convex setting and did not observe this issue.
> > > >
> > > > To acknowledge the merits of your study, I am increasing my score to 6.

---

> > > > > ### Author Response · Authors · 2024-08-09
> > > > > **Thanks for the review**
> > > > >
> > > > > **We first want to express our sincere gratitude for your kind ongoing discussion. From the review and discussion, we are also confident that your research in the area of convex problems is very profound.** We are honored to share some of the issues we are currently encountering in the non-convex experiments and the inconsistencies with the theory.
> > > > >
> > > > > Before the discussion stage, we carefully learned the series of papers you mentioned above. At the same time, we observed an issue: the studies in convex objective research are precise. Both the accuracy of local Lagrangian function solutions and the constants in the assumptions can be meticulously controlled. However, it is challenging to achieve effective control in experiments with non-convex objectives, especially as the models become more complex and the datasets grow larger. Therefore, there is a gap between theory and practice. This is why FedADMM has been theoretically proven to achieve partial participation, but still faces significant issues in practice, they have not rigorously verified whether the experiments can satisfy the assumptions and conditions, and how to satisfy them. This is also a challenge and difficulty in the current field of research on non-convex problems.
> > > > >
> > > > > In this paper, our motivation comes from extensive experimental validation. We found that divergence never occurs in the case of full participation. It only happens with partial participation, and the divergence becomes more severe as the proportion of partial participation decreases. We understand your concern: "unless too aggressive step sizes are chosen, but this is normal." We have explored nearly all possible hyperparameters to find stable convergence scenarios (as shown in the appendix). Unfortunately, divergence at low participation rates is difficult to control. Combined with the descriptions of this experimental phenomenon in many previous works, we are confident that there is an important issue waiting to be addressed here. This also inspired us to propose a technique that makes partial participation resemble full participation as closely as possible.
> > > > >
> > > > > **We appreciate your recognition of our paper.** We will revise the paper according to the results of our discussions:
> > > > >
> > > > > (1) adding an introduction to the theoretical advances of federated dual methods in section related work;
> > > > >
> > > > > (2) correcting the $O(L^3)$ notation;
> > > > >
> > > > > (3) fixing all typos;
> > > > >
> > > > > (4) incorporating some paragraphs from the discussion into the main text to further clarify the situations in which the problem arises.
> > > > >
> > > > > **Discussion with you has been very helpful and has made the paper more complete. We thank you again for your time and effort in reviewing this paper. If you have any further questions, we are very happy to continue discussing these academic issues with you and learn from you.**

---

### Official Review · Reviewer_b9A4 · 2024-07-09

**Soundness:** 3
**Presentation:** 3
**Contribution:** 3
**Rating:** 7
**Confidence:** 5

**Summary:**

This paper investigates the issue of dual drift caused by the mismatch between primal and dual variables when using partially participated training in federated learning (FL). It proposes a novel method, A-FedPD, which employs virtual dual updates to mitigate these negative impacts. Comprehensive theoretical analysis and extensive experiments are presented to verify the effectiveness of the proposed method.

**Strengths:**

1. The experiment in Figure 1 vividly illustrates the source of defects in primal dual methods under partial participation.
2. The study presented in this paper is novel. Federated primal dual methods have long exhibited varying degrees of instability in experiments. The analysis based on dual drift provided in this paper is robust and useful.
3. The quadratic term in the primal dual method is used in this paper to obtain a reduced form of the stability bound in the iterative process, ultimately achieving a constant bound in Equation (10). This demonstrates that the proposed method is superior to the federated averaging method in terms of generalization under the same training process. This conclusion is broad and can be extended to other primal dual methods.
4. The experiments were solid and comprehensive in their analysis. In terms of both communication efficiency and wall-clock time testing, the A-FedPD method shows strong performance.

**Weaknesses:**

1. What is the variant A-FedPDSAM? I didn’t see an introduction to this algorithm in the text; please remind me if I missed this part.
2. The performance of the algorithm is improved by trading space for time. During a global update, storing the dual variables requires considerable resources. Although federated learning primarily considers communication bottlenecks, this still increases server-side consumption.
3. I noticed that the FedDyn column in the main table indicates failure. Were all hyperparameters tested in the experiment? There should be an additional discussion to clarify whether this failure is a special case or not.

**Questions:**

The same with the weaknesses part.

**Limitations:**

The same with the weaknesses part.

---

> ### Author Rebuttal · Authors · 2024-08-04
>
> **W1: What is the variant A-FedPDSAM? I didn’t see an introduction to this algorithm in the text; please remind me if I missed this part.**
>
> Due to space constraints in the main text, we have noted at the bottom of page seven (in the first paragraph where the Experiments section begins) that the method A-FedPDSAM is introduced in appendix A.1.
>
> Actually, A-FedPDSAM is a straightforward but useful variant to improve the generalization efficiency in the experiments. The vanilla A-FedPD adopts the SGD as the local optimizer to solve the local lagrangian objective. While in A-FedPDSAM, we use SAM [X1] as the local optimizer to solve the local lagrangian objective. The design of SAM can provide better generalization performance. We list the main differences in the following.
>
> Local SGD optimizer:
> $$
> \theta_{i,k+1}^t=\theta_{i,k}^t - \eta^t(\nabla f_i(\theta_{i,k}^t,B) + \lambda_i + \rho(\theta_{i,k}^t - \theta^t)).
> $$
> Local SAM optimizer:
> $$
> \theta_{i,k+1}^t=\theta_{i,k}^t - \eta^t(\nabla f_i(\theta_{i,k}^t + \rho\frac{\nabla f_i(\theta_{i,k}^t,B)}{\Vert\nabla f_i(\theta_{i,k}^t,B)\Vert},B) + \lambda_i + \rho(\theta_{i,k}^t - \theta^t)).
> $$
> Other operations are essentially consistent with the A-FedPD method.
>
>
>
> **W2: The performance of the algorithm is improved by trading space for time. During a global update, storing the dual variables requires considerable resources. Although federated learning primarily considers communication bottlenecks, this still increases server-side consumption.**
>
> Thank you very much for pointing this out. As we discussed in the limitations, although our proposed method achieves state-of-the-art (SOTA) performance, it could be further improved in scenarios with a very large client scale, such as cross-device federated settings. Our paper primarily reviews and summarizes the issue of experimental non-convergence caused by increased model scale and more complex datasets in FedADMM-like methods. Then we organize this issue from the perspective of symmetry between the primal and dual problems as dual drift. Virtual dual updates are a simple and effective solution, and the storage cost is acceptable in cross-silo federated scenarios. We also look forward to further researching more efficient techniques for cross-device federated scenarios in the future.
>
>
> **W3: I noticed that the FedDyn column in the main table indicates failure. Were all hyperparameters tested in the experiment? There should be an additional discussion to clarify whether this failure is a special case or not.**
>
> We conducted all hyperparameter searching as shown in Table 3 (page 21) and tried almost all possible hyperparameter combinations. In fact, we also observed that while FedDyn performs reasonably well in CIFAR-10 experiments, it has a very high requirement for learning rate decay, necessitating a very slow learning rate decay. However, on CIFAR-100 (where the model structure and tasks become more complex), its stability drops sharply, requiring continuous reduction of the penalty term in the Lagrangian function to maintain stability. This observation inspired us to consider whether the asymmetry between the primal and dual objectives is causing this instability. When the penalty coefficient decreases to nearly zero, the dual variables almost stop updating and remain the initial zero vector, causing the entire Lagrangian function to nearly degenerate into the original function $f_i(\theta)$. This phenomenon can also be observed in the experiments reported in [X1]. We also encountered similar issues in the FedADMM experiments. This is not a coincidence or an anomaly, but a genuine issue.
>
> [X1] Xu J, Wang S, Wang L, et al. Fedcm: Federated learning with client-level momentum[J]. arXiv preprint arXiv:2106.10874, 2021.
>
> Thank you again for reading our rebuttal. We are also honored to have this discussion with you, which has made our submission more complete. If you have any further questions or issues to address, we would be very happy to continue discussing them with you.

---

> > ### Comment · Reviewer_b9A4 · 2024-08-08
> >
> > Thanks for the authors' response. I have no additional questions concerning this paper.

---

> > > ### Author Response · Authors · 2024-08-08
> > > **Thanks for the review**
> > >
> > > Thank you again for the time and effort paid for reviewing our submission. We will revise the corresponding content based on the summary in the rebuttal.

---

### Official Review · Reviewer_YYQB · 2024-07-10

**Soundness:** 3
**Presentation:** 3
**Contribution:** 3
**Rating:** 5
**Confidence:** 3

**Summary:**

A primal-dual-based federated learning algorithm (A-FedPD) is proposed to mitigate the drift of local dual variables. In federated learning with partial participation training, the local dual variables in inactive clients can be drifted. To mitigate this issue, the propose A-FedPD would be effective since the local dual variables in unparticipated clients are aligned with the global average models. Through extensive experiments, including nonconvex neural network models, the effectiveness of the A-FedPD was empirically investigated.

**Strengths:**

(S1) Simple but effective method: It is well known that the local dual variables in unparticipated clients can be drifted in the primal-dual based federated learning algorithms. The idea of modifying the unparticipated local dual variables to bring them closer to the global model is simple but effective.

(S2) The proposed Algorithm 1 has been theoretically analyzed under certain assumptions, demonstrating a near-optimal convergence rate. One point of dissatisfaction is the comparison of convergence rates with several primal-dual based methods that will not be affected by dual drift in Table 7. Such an important results are to be included in the main paper.

(S3) The effectiveness of the proposed method has been validated through several image classification benchmark tests using neural network models. Its robustness against various stresses (participation ratio, local interval, data heterogeneity) has also been empirically investigated.

**Weaknesses:**

(W1) In the experiments of Sec. 5.2, there seems to be no significant difference between the proposed method FEDPD and the comparison method FEDSPEED. The reviewer has serious concerns about this point. Without introducing SAM, can the superiority not be empirically demonstrated?

(W2) In addition, the consideration of comparison methods in Sec. 5.2 seems insufficient. FEDSPEED performs better, but why is this? It also compares with primal methods such as SCAFFOLD and FedSAM, but why is the proposed method better than these methods?

(W3) Some relevant literature might be missing.

[a] R. Pathak et al., “FedSplit: An algorithmic framework for fast federated optimization,” NeurIPS 2020.

[b] G. Zhang et al., “Revisiting the Primal-Dual Method of Multipliers for Optimisation Over Centralised Networks”, IEEE Transaction on Signal and Information Processing, pp. 228-243, 2022.

**Questions:**

(Q1) Could you provide proof sketch for theoretical analysis in Sec. 4? I would particularly like to know about the novel points in comparison with analyses in prior studies, such as FedPD.

(Q2) On line 309, it is written that “we freeze most of the hyperparameters for all methods,” however these were searched in the list written in the Grid Search column of Table 3, right? I think it would be better to clearly write in the main paper that a hyperparameter search was conducted (see Appendix).

(Q3) In Figure 2(b), it seems that accuracy is lower when the local interval is short. Does the total number of updates differ according to the local interval?

**Limitations:**

Limitations are noted in the Appendix. Social impact is not mentioned.

---

> ### Author Rebuttal · Authors · 2024-08-04
>
> **W1: The reviewer has serious concerns about this point. Without introducing SAM, can the superiority not be empirically demonstrated?**
>
> FedSpeed method essentially uses **a variant of the local SAM optimizer.** To illustrate this, we can recall its implementation:
> $$
> g_{i,k,1}^t = \nabla F_i(x_{i,k}^t,B),
> $$
> $$
> y_{i,k}^t = x_{i,k}^t + \rho g_{i,k,1}^t,
> $$
> $$
> g_{i,k,2}^t = \nabla F_i(y_{i,k}^t, B),
> $$
> $$
> g_{i,k}^t = (1-\alpha)g_{i,k,1}^t + \alpha g_{i,k,2}^t.
> $$
> The ascent learning rate is set as $\rho=\frac{\rho_0}{\Vert\nabla F_i\Vert}$. Then we have:
> $$
> g_{i,k}^t = (1-\alpha)\nabla F_i(x_{i,k}^t,B) + \alpha\nabla F_i(x_{i,k}^t + \rho_0\frac{\nabla F_i(x_{i,k}^t,B)}{\Vert\nabla F_i(x_{i,k}^t,B)\Vert}, B).
> $$
> **It is a combination of the SGD gradient and the SAM gradient, the second part is actually the SAM gradient.** Therefore for a fair comparison, from the perspective of local optimizers, the corresponding version of FedPD/FedADMM/FedDyn is A-FedPD, while the corresponding version of FedSpeed is A-FedPDSAM (both of them adopt SAM-based optimizer in the local client). As observed by the reviewer, our method makes the performance of SGD-based optimizers approach that of SAM-based optimizers, which is a significant improvement.
>
> **W2: FEDSPEED performs better, but why is this? It also compares with primal methods such as SCAFFOLD and FedSAM, but why is the proposed method better than these methods?**
>
> Thank you for pointing out this. In W1, we have already explained above why the FedSpeed method is superior to A-FedPD, primarily because its local optimizer is essentially a variant of SAM. For a fair comparison, it should be compared with the variant A-FedPDSAM.
>
> As for the second question, when the FedDyn method was proposed [X1], it already summarized that due to the correction of the dual variable, **local models will converge to the global model.** This means that the primal-dual methods will adjust the local heterogeneous objective to ultimately align with the global objective. Our proposed method is primarily designed to address the training instability issues and some inherent shortcomings in primal-dual methods. For clearer comparisons, we classify some methods and list some results from Table 3.
>
> |    | type  | local optimizer | Acc in Dir-1.0 | Acc in Dir-0.1 |
> |  :----:  | :----:  | :----:  | :----:  | :----:  |
> | SCAFFOLD  | primal | SGD | 83.61 | 78.66 |
> | FedDyn  | primal-dual  | SGD | 84.20 | 79.51 |
> | A-FedPD | primal-dual  | SGD | 84.94 | 80.28 |
> | FedSpeed | primal-dual  | SAM | 85.11 | 80.86 |
> | A-FedPDSAM  | primal-dual  | SAM | 85.90 | 81.96 |
>
> [X1] Acar D A E, Zhao Y, Navarro R M, et al. Federated learning based on dynamic regularization[J]. arXiv preprint arXiv:2111.04263, 2021.
>
> **W3: Some relevant literature might be missing.**
>
> Thank you for pointing out these two important works. We will add them in the Related Work.
>
> **Q1: Could you provide proof sketch for theoretical analysis in Sec. 4? I would particularly like to know about the novel points in comparison with analyses in prior studies, such as FedPD.**
>
> (1) For optimization: [X2] provides classical proofs by independently bounding each term in the smoothness inequality, ultimately forming the final convergence conclusion. However, due to the complexity of the intermediate variables and relationships that need to be analyzed, the proof introduces a large number of constants. Then FedDyn employs a more refined technique [X1]. Instead of independently bounding each term, the proof in FedDyn combines the global update gaps and the local update gaps by a specific coefficient, yielding an arithmetic sequence. However, proofs in FedDyn must rely on the assumption of the local exact solution, i.e. $\nabla L_i=0$. Clearly, this is overly idealized, as optimization will always introduce some errors. Therefore, to make the proof process more general, we adapt the assumption of gradient error from [X2], i.e. $\nabla L_i=e$ where $e$ is treated as a bounded error. Our proof extends the simplified version of FedDyn to a more general version that allows for local inexact solutions and re-evaluates the necessary conditions for constructing the combined terms.
>
> (2) For generalization: **To our knowledge, current works have not provided generalization analysis for federated primal-dual methods.** We provide error bounds based on stability analysis here and prove that the generalization error bound of our proposed A-FedPD is lower than the FedAvg method. The main property is reflected through Eq.(10) in our paper. The local updates in primal-dual methods can be decayed by a coefficient less than 1 ($1-\eta^t\rho$), which implies fewer stability errors compared with the FedAvg method.
>
> [X1] Acar D A E, Zhao Y, Navarro R M, et al. Federated learning based on dynamic regularization[J]. arXiv preprint arXiv:2111.04263, 2021.
>
> [X2] Zhang X, Hong M, Dhople S, et al. Fedpd: A federated learning framework with adaptivity to non-iid data[J]. IEEE Transactions on Signal Processing, 2021, 69: 6055-6070.
>
> **Q2: I think it would be better to clearly write in the main paper that a hyperparameter search was conducted (see Appendix).**
>
> Thank you for pointing out this. We will add the sentence "Detailed hyperparameter search was conducted (see Appendix A.2)" in section 5.1.
>
>
> **Q3: In Figure 2(b), it seems that accuracy is lower when the local interval is short. Does the total number of updates differ according to the local interval?**
>
> Thank you for this question. In the experiments of Figure 2 (b), we fix the communication rounds as $800$ and change the local intervals from $[10,20,50,100,200]$. Since the learning rate is decayed by each communication round, we must ensure that the learning rate is reduced to the same after all experiments are finished. We will further clarify the experimental setup.
>
> Thank you again for reading our rebuttal. If you have any further questions, we would be very happy to continue the discussion with you.

---

> > ### Comment · Reviewer_YYQB · 2024-08-09
> >
> > W1: I clearly understand your explanation. However, I couldn't extract this claim from the original version. Could you improve the experimental section to explain this better?
> >
> > W2: Including this table that explains the relationships between the methods in the Appendix would help enhance the readers' understanding.
> >
> > Q1: Adding this to the Appendix would make it easier to understand the contributions in the theoretical analysis.
> >
> >
> > I do not have any questions furthermore. I keep my score.

---

> > > ### Author Response · Authors · 2024-08-09
> > > **Thanks for the review**
> > >
> > > We are very grateful for your review of our rebuttal and are pleased that all your concerns have been addressed.
> > >
> > > We greatly appreciate your suggestions. We will add a horizontal table above Table 2 in our paper to categorize each method, and add the explanations in the rebuttal above regarding optimization and generalization techniques in the appendix. Thank you again for your recognition of our work.

---

### Official Review · Reviewer_6Q4p · 2024-07-12

**Soundness:** 3
**Presentation:** 4
**Contribution:** 3
**Rating:** 7
**Confidence:** 4

**Summary:**

This paper studies the primal-dual-based FL algorithm with partial client participation. The inactiveness of the local clients causes both local primal and dual variables to drift from their expected value and slows down FedPD's convergence. This paper provides a fix to the FedPD algorithm in the partial participation setting by moving the dual update to the server side and allowing the server to store the dual variables. The theoretical result shows that A-FedPD can achieve the same convergence rate as FedPD ($O(1/T)$), and the error bound is better than the prima-only FL algorithm. Numerical results also show that the proposed algorithm achieves the best accuracy in different settings.

**Strengths:**

1. Soundness: the theoretical analysis provides a clear convergence and generalization result for the proposed algorithm. Numerical results also show that the proposed algorithm outperforms the existing algorithms on different models and data distributions.
2. Clarity: the paper is clearly written with adequate reference to prior works and clear notations and results.

**Weaknesses:**

1. Memory inefficiency: The algorithm requires saving the local dual variable of all clients. When C is large, this might cause a large memory cost on the server, especially in the FL setting. This might restrict the proposed algorithm's use case. This is discussed in the limitations.

**Questions:**

How would A-FedPD (and other algorithms) perform when the communication patterns of different clients are different, i.e., $\mathbb{E}_P[\theta_i] \neq \bar{\theta}$?

---

> ### Author Rebuttal · Authors · 2024-08-04
>
> **W1: The algorithm requires saving the local dual variable of all clients. When C is large, this might cause a large memory cost on the server, especially in the FL setting. This might restrict the proposed algorithm's use case. This is discussed in the limitations.**
>
> Thank you very much for pointing this out. As we discussed in the limitations, although our proposed method achieves state-of-the-art (SOTA) performance, it could be further improved in scenarios with a very large client scale, such as cross-device federated settings. Our paper primarily reviews and summarizes the issue of experimental non-stability caused by increased model scale and more complex datasets in FedADMM-like methods. Then we organize this issue between the primal and dual problems as dual drift. Virtual dual updates are a simple and effective solution, and the storage cost is acceptable in cross-silo federated scenarios. We also look forward to further researching more efficient techniques for cross-device federated scenarios in the future.
>
>
> **Q1: How would A-FedPD (and other algorithms) perform when the communication patterns of different clients are different, i.e., $\mathbb{E}_P[\theta_i]\neq\overline{\theta}$?**
>
> Thank you very much for raising this interesting question. First, we clarify that the vanilla definition of $\overline{\theta}$ is not the absolute average $\frac{1}{P}\sum \theta_i$, but a weighted average $\sum p_i \theta_i$ where $p_i$ corresponds to the importance. In most work, the global objective is set as $F(w)=\frac{1}{m}\sum f_i(\theta)$, yielding that the global consistency we are solving for is simplified on the absolute average form.
>
> Therefore, based on your question, the corresponding issue I understand is: (1) global objective is still $F(w)=\frac{1}{m}\sum f_i(\theta)$ so the global consensus is still the absolute average form $\overline{\theta}$; (2) due to certain algorithm designs, such as importance sampling or client dropout, the sampling probability $p_i$ for each client is no longer consistent with the original global objective. This generates a new gap between the true global objective and the constructed objective.
>
> This discussion requires some supporting assumptions, which we will briefly discuss here. From the perspective of optimization, the constructed objective should play a similar role of the vanilla global objective which can guarantee the training process converges. Under this condition, the gap between $\mathbb{E}\Vert\nabla F(\mathbb{E}_P[\theta_i])\Vert$ and $\mathbb{E}\Vert\nabla F(\overline{\theta})\Vert$ must be upper bounded and can deminished to zero. We refer to this as "different but bounded". We have not thoroughly derived the impact that the distribution $P$ might have on the optimization results. However, given that the optimization results still converge, we can assume that the gap here remains bounded during training. Thus, adopting $\overline{\theta}$ is still applicable, but with an additional error term at the final convergence rate.
>
> There is another scenario that corresponds to this, which is the asynchronous setup. This situation appears to be more complex, but we believe we can draw on some ideas from asynchronous distributed ADMM methods [X1, X2]. Some related techniques like "partial barrier" and "bounded delay" can be introduced to mitigate the risk of unstable updates. The core idea is to specify the error between the global objective and global consensus, and then consider further optimization of this error.
>
> We think this would be a good academic perspective, and indeed a topic worth exploring further in the future. To further answer this question in detail, we may need more conditions and problem definitions for a more specific analysis. We hope that our current response addresses your concerns.
>
> [X1] Zhang R, Kwok J. Asynchronous distributed ADMM for consensus optimization[C]//International conference on machine learning. PMLR, 2014: 1701-1709.
>
> [X2] Hong M. A distributed, asynchronous, and incremental algorithm for nonconvex optimization: An ADMM approach[J]. IEEE Transactions on Control of Network Systems, 2017, 5(3): 935-945.
>
>
> Thank you again for reading our rebuttal. If you have any further questions, we would be very happy to continue the discussion, as it will help further refine our submission.

---

> > ### Comment · Reviewer_6Q4p · 2024-08-07
> >
> > Thanks for the response. I don't have any further questions.

---

> > > ### Author Response · Authors · 2024-08-08
> > > **Thanks for the review**
> > >
> > > Thank you again for the time and effort paid for reviewing our submission.

---

### Author Rebuttal · Authors · 2024-08-05

**We thank the four reviewers for their valuable time and effort in reviewing our submission.** Reviewers 6Q4p, b9A4, and YYQB provided positive feedback with rates of 7, 7, and 5, respectively. After reading all the review comments, we noticed that the response from reviewer c2hH mentioned some concerns regarding the main contributions and research objectives of this paper, such as the 'definition of dual drift' and 'whether it should be named as dual drift'. Given that this is the main issue studied in this paper, we hope to provide a comprehensive summary in this overall response to clarify misunderstandings.

**A. Main Contributions**

(1) We want to emphasize that the target of this paper is neither to propose a better theoretical framework to surpass the SOTA convergence rate nor to explore how federated learning can achieve lower communication bits. This paper investigates the training instability phenomenon widely observed in some classic federated primal-dual methods (we name it the "dual drift" issue) in experiments and proposes a simple and effective solution to avoid this issue.

(2) The optimization proof is provided to demonstrate that our proposed technique does not cause the primal-dual methods to diverge. Meanwhile, it can significantly improve the test performance in experiments.

(3) To the best of our knowledge, this paper is the first to provide a generalization error based on stability analysis for federated primal-dual methods. We explain the superior stability over the classical FedAvg method from the perspective of local training stability.

**B. Dual Drift**

We appreciate the reviewer c2hH for listing a series of works to demonstrate that partial participation in primal-dual methods is theoretically feasible. We have never claimed in this paper that they are theoretically infeasible. In fact, the FedADMM method, which exhibits training instability as shown in our Figure 1, is also theoretically proven to support partial participation [X1]. **However, we have verified that this method encounters significant challenges in experiments, with severe fluctuations as the participation rate decreases.** We believe that theoretical proofs are one of the assurances of an algorithm's validity, but they are not everything. In practice, various detailed issues still need to be effectively addressed.

Based on extensive experiments and analysis, we attribute the cause of this instability phenomenon as follows. Methods like ADMM require alternating updates of the primal and dual variables to ensure that the Lagrangian function eventually converges to the original objective. Under the low participation ratio, a client waits for a long time to be activated. For the local client, it is as if the primal variables are continuously being updated, while the dual variables remain in a stagnant state. When it is suddenly activated, the lagging dual variables result in the current primal variables being in a very poor initialization state for the local Lagrangian function. Therefore, we have named this phenomenon ``dual drift". Our solution is to perform virtual dual updates on the dual variables based on estimated primal variables for those unparticipated clients, to ensure they do not lag too far behind.

[X1] Wang H, Marella S, Anderson J. Fedadmm: A federated primal-dual algorithm allowing partial participation[C]//2022 IEEE 61st Conference on Decision and Control (CDC). IEEE, 2022: 287-294.


**C. A series of works [X1-X5] mentioned by Reviewer c2hH**

We are very grateful to reviewer c2hH for introducing us to this series of excellent theoretical works [X1-X5]. We will add a section on the theoretical advancements of federated primal-dual methods in the related work section to introduce them. However, as mentioned above, the goal of this paper is not to propose an optimization convergence analysis that surpasses existing methods but rather to address the currently widespread training instability phenomenon in experiments.

**We need to emphasize that this phenomenon is not easily observed on small datasets and small models.** For example, FedDyn may perform normally with the ResNet-18 model on the CIFAR-10 dataset but shows unstable training on CIFAR-100. When the model size is further increased, e.g. Transformers, this phenomenon becomes even more pronounced. [X1-X5] primarily focus on the progress of optimization proofs, with experiments mostly centered around the tiny logistic regression models and smaller libsvm datasets. Additionally, the experimental scales they used are typically smaller than 50. The focus of the work's analysis scenarios and experimental setups differs from ours significantly, which makes it difficult for us to make a unified comparison.

[X1] Grudzień M, Malinovsky G, Richtárik P. Can 5th generation local training methods support client sampling? yes![C]//International Conference on Artificial Intelligence and Statistics. PMLR, 2023: 1055-1092.

[X2] Grudzień M, Malinovsky G, Richtárik P. Improving accelerated federated learning with compression and importance sampling[J]. arXiv preprint arXiv:2306.03240, 2023.

[X3] Mishchenko K, Malinovsky G, Stich S, et al. Proxskip: Yes! local gradient steps provably lead to communication acceleration! finally![C]//International Conference on Machine Learning. PMLR, 2022: 15750-15769.

[X4] Condat L, Richtárik P. RandProx: Primal-dual optimization algorithms with randomized proximal updates[J]. arXiv preprint arXiv:2207.12891, 2022.

[X5] Condat L P, Malinovsky G, Richtárik P. Tamuna: Accelerated federated learning with local training and partial participation[J]. 2023.

Thank all reviewers for reading our rebuttal. We hope this summary clarifies the main research objectives and contributions of this paper. Responses to other concerns are provided in separate replies. If there are any unresolved concerns, we are more than happy to engage in further discussion with the reviewers.

---

### Decision · Program_Chairs · 2024-09-25

**Decision:**

Accept (spotlight)

**Comment:**

This paper studies primal-dual federated learning algorithms. It demonstrates that primal-dual methods can suffer from a so-called "drift" in the dual variables of inactive clients on non-convex problems. The paper proposes updating the dual variables of non-participating clients via global average models to mitigate the drift. The effectiveness of the approach is demonstrated through numerical experiments and is theoretically verified under smoothness and Lipschitz assumptions.

The dual drift phenomenon does not exist in the convex setting, for which most of these methods were originally developed. The reviewers remarked that the differences between these cases should be discussed more prominently in the work, and that the findings and phenomena should be contrasted with existing theoretical work on the convex setting. The changes proposed by the authors during the lengthy discussion with the reviewers appear sufficient if they can be integrated into the revision.

The reviewing committee finds that the paper has the merit of stimulating discussion on this important topic and can advance research on practical federated learning algorithms.